# STAGE: A Foundation Model for Spatial Transcriptomics Analysis via Graph Embeddings with Hierarchical Prototypes

## Abstract

Spatial transcriptomics offers an unprecedented opportunity to elucidate the spatial organization of tissues by capturing gene expression profiles while preserving tissue architecture. This enables the identification of spatial niches and deepens our understanding of tissue function and disease-associated microenvironments. However, consistent identification of spatial domains across samples, tissues, and even technological platforms remains a formidable challenge, due to low-dimensional and heterogeneous gene panels across platforms, pronounced batch effects, and substantial biological variability between samples. To address these limitations, we propose STAGE, a generalizable foundation model for spatial transcriptomics via graph embeddings. At its core, STAGE introduces a hierarchical prototype mechanism to capture global semantic representations of spatial niches, alongside an efficient online expectation-maximization algorithm to enable scalable learning from large-scale heterogeneous data. Pretrained on a large dataset comprising 32 million cells from 18 tissue types, STAGE learns robust cell representations within their neighborhood graphs and supports niche inference for domain recognition. Comprehensive evaluations on multiple benchmark datasets demonstrate that STAGE substantially enhances domain consistency in cross-platform, cross-sample, and cross-tissue spatial domain identification tasks, outperforming existing state-of-the-art methods. Furthermore, STAGE supports critical downstream biological analyses, highlighting its strong potential as a powerful tool in biological research.

## 1 Introduction

Spatial transcriptomics (ST) has fundamentally transformed biological research by enabling high-resolution localization of gene expression within intact tissue microenvironments (Bressan et al., 2023). By retaining spatial context, ST supports the delineation of tissue architecture, identification of spatial domains (or "niches"), and elucidation of microenvironmental organization underlying physiological functions and disease mechanisms (Rao et al., 2021).

The rise of diverse ST platforms—ranging from whole-transcriptome sequencing to targeted gene panels—has introduced substantial heterogeneity in spatial resolution, sequencing depth, gene coverage, and tissue origin. ST technologies are broadly categorized into two types: sequencing-based methods such as next-generation sequencing (NGS), which provide genome-wide expression profiling but limited spatial resolution (McCombie et al., 2019); and imaging-based approaches, including Xenium (Janesick et al., 2023), MERSCOPE (Chen et al., 2015), and CosMX (He et al., 2022), which achieve subcellular resolution but are constrained by the number of detectable genes. Although imaging platforms increasingly support larger and customizable gene panels—including dynamic substitution of low-quality genes during experimentation—this flexibility poses new challenges for cross-platform integration and standardization (Figure 1 (a)).

In multi-section analyses, pronounced batch effects often obscure shared spatial patterns and impede cross-sample comparisons. Without effective correction, such artifacts hinder the discovery of consistent biological principles (Figure 1 (b) and Figure 1 (c)).

Identifying spatial domains remains a core task in ST analysis (Zormpas et al., 2023). Methods such as Louvain (Blondel et al., 2008), Leiden (Traag et al., 2019), STAGATE (Dong & Zhang, 2022), GraphST and (Long et al., 2023) perform well on spot-level data from NGS platforms. However, they face key limitations: (1) Gene panel dependency—reliance on preselected gene sets reduces adaptability across platforms and resolutions (Xu et al., 2022); (2) Per-sample retraining—most require retraining and tuning for each dataset, limiting generalizability beyond specific tissues or platforms (Shen et al., 2024); and (3) Lack of unified semantic representation—conventional clustering methods focus on local structures, impeding modeling of higher-level spatial semantics and reducing biological interpretability (Hu et al., 2021).

Recently, foundation models such as scGPT (Cui et al., 2024) and Novae (Blampey et al., 2024) have emerged. scGPT leverages large-scale single-cell transcriptomes to reduce reliance on limited spatial gene panels, enhancing cross-platform generalization—albeit with potential compromise of spatial specificity due to non-spatial signals. Novae, a graph-based self-supervised framework, introduces spatial embedding alignment to improve correspondence across tissues. Nonetheless, these models still focus mainly on local neighborhoods and lack mechanisms to capture global, multi-scale spatial patterns, limiting their ability to model cross-domain and cross-tissue semantics.

To address these fundamental challenges in spatial transcriptomics analysis, we propose **STAGE** (**S**patial **T**ranscriptomics **A**nalysis via **G**raph **E**mbeddings with Hierarchical Prototypes), a generalizable foundation model that enables robust and consistent spatial domain identification across diverse experimental conditions (the overall framework is illustrated in Figure 1 (d) and Figure 1 (e)). STAGE makes four key contributions: **(1) Universal Clustering Framework:** STAGE enables consistent spatial clustering across samples, technologies and tissues without sample-specific optimization, standardizing spatial transcriptomics analysis; **(2) Self-supervised Learning:** Combining swapped contrastive learning with online EM optimization, our approach trains on unlabeled ST data at scale; **(3) Comprehensive Validation:** Extensive experiments demonstrate STAGE's versatility across diverse tasks, establishing it as a universal analysis tool.

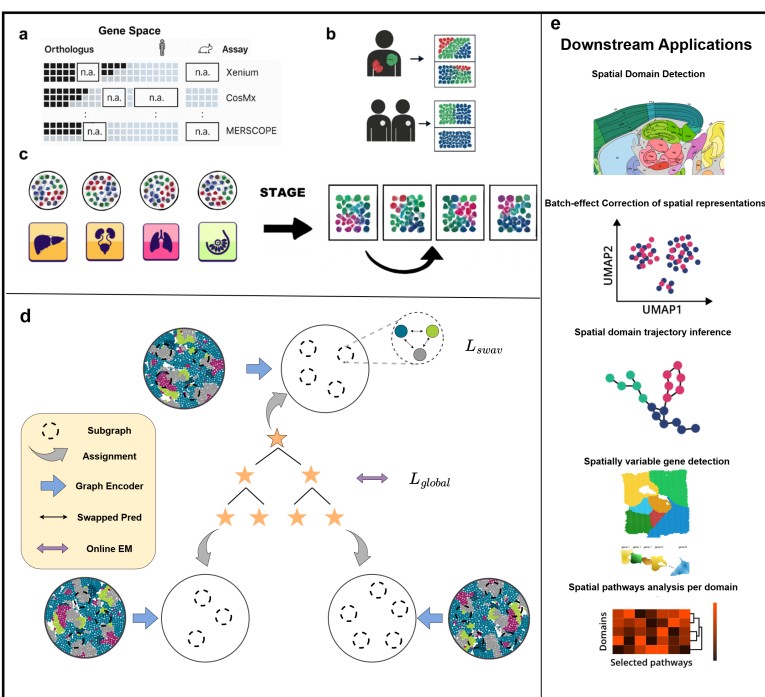

Figure 1: **(a)** Gene panel limitations cause low-dimensional, inconsistent features, leading to batch effects; **(b)** Biological heterogeneity; **(c)** STAGE achieves cross-platform, cross-sample, and cross-tissue integration, revealing shared spatial domains; **(d)** An overview of STAGE; STAGE introduces a hierarchical prototype mechanism to capture global semantic representations of spatial niches, alongside an efficient online expectation-maximization algorithm to enable scalable learning from large-scale heterogeneous data; **(e)** Downstream Applications.

## 2 RELATED WORK

### 2.1 CLASSICAL SPATIAL CLUSTERING

Early spatial domain detection methods typically construct weighted graphs by combining gene expression and spatial proximity, followed by community detection. The Louvain algorithm, though efficient, often yields fragmented domains with poor spatial continuity. Leiden improves connectivity via a two-level node–edge refinement but remains sensitive to batch effects in multi-section integration. Graph neural networks (GNNs) effectively model gene expression and spatial structure for domain identification. STAGATE integrates expression similarity via graph attention to enable adaptive edge weighting and 3D alignment. GraphST applies contrastive learning to align local and global patterns across sections without explicit batch correction. Despite strong performance, GNN methods often rely on dataset-specific hyperparameters, limiting their cross-platform generalization.

### 2.2 PRETRAINED FOUNDATION MODELS

To address the generalization limits of traditional methods, pretrained foundation models like scGPT and Novae have emerged to enable unified representation learning across platforms and tissues. scGPT, inspired by language modeling, is pretrained on large-scale single-cell data via masked-token prediction to learn transferable gene–gene and cell–gene relationships. Novae, a self-supervised graph model, introduces spatial embedding alignment loss for correcting platform discrepancies and supports zero-shot spatial domain prediction with integrated batch correction.

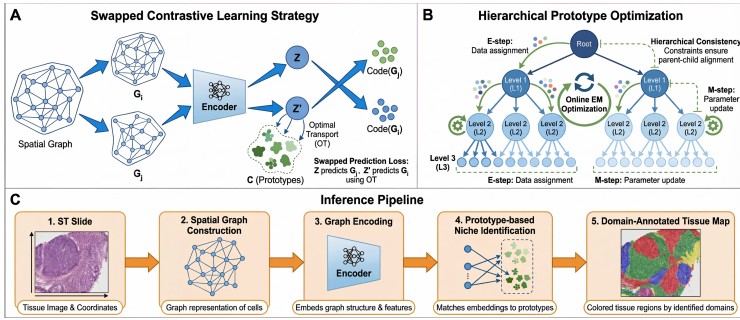

Figure 2: **(a) SwAV contrastive learning.** Two augmented graph views generate embeddings Q and Q', which are assigned to prototypes via optimal transport (OT) with swapped contrastive loss; **(b) Online EM optimization.** Hierarchical prototypes tree is updated through online expectation-maximization algorithm for scalable learning from ST data; **(c) STAGE inference pipeline.** Input ST slide is processed through spatial graph construction, graph encoding, and prototype-based spatial niche identification.

## 3 METHODOLOGY

### 3.1 PROBLEM FORMULATION AND PRELIMINARIES

We consider an omic set $O$ in the complete vocabulary (e.g., human or mouse genome) and denote $N_o$ as the omic set size. A spatial transcriptomics slide captures expression profiles of a subset $P \subseteq \{1, \ldots, N_o\}$, representing the gene panel indices for this specific technology. The expression data is represented as $X = (x_1, \ldots, x_N) \in \mathbb{R}^{N \times |P|}$, where $N$ denotes the number of cells and $|P|$ represents the panel size. This matrix contains normalized and log-transformed gene expression values.

**Cross-platform Challenges.** Different spatial transcriptomics technologies exhibit significant heterogeneity: gene panels vary dramatically ($P_i \neq P_j$ for platforms $i$ and $j$), batch effects introduce systematic biases, and biological variability exists across samples and tissues. These challenges necessitate robust representation learning that can generalize across diverse experimental conditions while preserving meaningful spatial organization.

## 3.2 STAGE FRAMEWORK OVERVIEW

STAGE addresses these challenges through a unified self-supervised learning framework that simultaneously models local spatial relationships and global tissue organization. The architecture consists of three synergistic components:

**Gene Embedder.** Maps heterogeneous gene panels to a unified embedding space, enabling cross-platform compatibility through learnable gene representations initialized with biological priors.

**GAT Encoder.** Learns spatial representations from multi-scale neighborhood graphs using attention mechanisms to capture both local cellular interactions and broader spatial context. More specifically, for a given cell $i$, we define a subgraph $G_i$ consisting of cells within $n_{\text{local}}$ hops, where node features are the gene expression vectors of corresponding cells.

**Hierarchical Prototype Head.** Combines SwAV's contrastive learning with our novel OT-enhanced online EM algorithm to discover hierarchical spatial organization, from fine-grained cellular niches to tissue-level domains.

This integrated design enables STAGE to learn representations that are both locally coherent and globally structured.

## 3.3 SPATIAL NEIGHBORHOOD CONSTRUCTION

Following standard practice in spatial transcriptomics, we construct cell graphs based on Euclidean distances between spatial coordinates. For each cell $v$, we define multi-scale neighborhoods:

$$\mathcal{N}_L(v) = \{u : d(u,v) \leq r_{\text{local}}\}, \tag{1}$$
$$\mathcal{N}_V(v) = \{u : r_{\text{local}} < d(u,v) \leq r_{\text{view}}\}, \tag{2}$$

where $d(u,v)$ denotes the Euclidean distance. This formulation jointly captures immediate microenvironments and broader tissue structures, enabling analysis across biological scales.

## 3.4 LOCAL SPATIAL STRUCTURE LEARNING

We extend the SwAV framework (Caron et al., 2020) to spatial transcriptomics, introducing a contrastive scheme that aligns representations of spatially proximal regions. Unlike traditional methods, our formulation eliminates costly pairwise comparisons while preserving their discriminative power, enabling efficient learning of local spatial structure.

### 3.4.1 OPTIMAL TRANSPORT FOR BALANCED CLUSTER ASSIGNMENT.

The core challenge in contrastive learning is preventing representation collapse while ensuring meaningful cluster formation. We address this through optimal transport, which provides a principled approach to balanced assignment.

Given a batch of embeddings $Z \in \mathbb{R}^{B \times d}$ and prototype matrix $C \in \mathbb{R}^{K \times d}$, we solve the entropic optimal transport problem:

$$Q^* = \arg\min_{Q \in \mathcal{Q}} \langle Q, -CZ^T \rangle + \epsilon H(Q) \tag{3}$$

where $H(Q) = -\sum_{i,j} Q_{ij} \log Q_{ij}$ is the Shannon entropy, $\epsilon > 0$ is the regularization parameter, and $\mathcal{Q}$ is the transportation polytope, $\mathcal{Q} = \left\{ Q \in \mathbb{R}_+^{B \times K} : Q\mathbf{1}_K = \frac{1}{B}\mathbf{1}_B, Q^T\mathbf{1}_B = \frac{1}{K}\mathbf{1}_K \right\}$.

This formulation ensures that: (1) each sample is assigned to exactly one prototype on average, (2) each prototype receives equal assignment mass, and (3) the assignment is smooth due to entropy regularization.

### 3.4.2 SWAPPED PREDICTION LOSS.

Given two spatially related subgraphs $(G_i, G_j)$ separated by $n_{\text{view}}$ edges, we obtain embeddings $z_i$ and $z_j$. The SwAV loss aligns their cluster assignments:

$$\mathcal{L}_{\text{swav}} = -\frac{1}{2} \sum_{(i,j)} \sum_{k=1}^{K} \left( q_{ik} \log p_{jk} + q_{jk} \log p_{ik} \right), \tag{4}$$

where $q_i = Q_i^*$ is the balanced assignment from optimal transport and $p_i = \text{softmax}(z_i^\top C/\tau)$ is the prototype distribution with temperature $\tau$.

By swapping predictions across neighboring subgraphs, this loss enforces representation consistency without negative sampling. This property is especially advantageous for spatial transcriptomics, where dense local interactions make conventional contrastive objectives inefficient.

### 3.5 GLOBAL SPATIAL SEMANTIC LEARNING VIA HIERARCHICAL PROTOTYPES

Local structure learning captures short-range interactions, but tissues exhibit multi-scale organization from cellular niches to anatomical regions. To model this, we introduce a hierarchical prototype framework coupled with an OT-enhanced online EM algorithm.

#### 3.5.1 TREE-STRUCTURED PROTOTYPE HIERARCHY.

We arrange prototypes in a tree $\mathcal{T} = \{\mathcal{C}^l\}_{l=1}^{L_p}$, where each level encodes spatial patterns at a distinct resolution:

$$\mathcal{C}^l = \{c_1^l, \ldots, c_{K_l}^l\}, \quad c_i^l \in \mathbb{R}^d. \tag{5}$$

Each prototype $c_i^l$ has children in $\mathcal{C}^{l+1}$ and a unique parent, enforcing a valid tree structure with at least two children per non-leaf node.

This hierarchy reflects the biological reality of spatial transcriptomics: fine-grained cellular states at lower levels are recursively aggregated into broader tissue domains at higher levels. Unlike flat prototypes, our tree-structured organization explicitly encodes biological hierarchy, enabling global semantic representation across scales.

#### 3.5.2 OT-ENHANCED ONLINE EXPECTATION-MAXIMIZATION.

We cast hierarchical prototype learning as a latent-variable model, where each spatial graph $G$ follows a path $z_G = \{z_G^1, \ldots, z_G^{L_p}\}$ through the prototype tree, with the constraint $z_G^{l+1} \in Child(z_G^l)$ ensuring hierarchical consistency.

Our central innovation integrates optimal transport into the EM framework, providing balanced assignments while maintaining hierarchical consistency:

**E-step (OT-enhanced Assignment).** Latent assignments are sampled top-down along the tree:

$$z_G^1 \sim \text{Cat}(\text{softmax}(h_G^\top \mathcal{C}^1/\tau)), \tag{6}$$

$$z_G^{l+1} \sim \text{Cat}(\text{softmax}(h_G^\top Child(z_G^l)/\tau)). \tag{7}$$

At each level $l$, Sinkhorn-Knopp yields balanced transport plans:

$$Q_l^* = \text{Sinkhorn}(C^l Z^T/\epsilon_l) \text{ subject to } z_G^{l+1} \in Child(z_G^l). \tag{8}$$

This constraint ensures that the hierarchical structure is preserved during assignment, preventing inconsistent paths through the tree.

**M-step (Hierarchical Parameter Update).** update network parameters $\theta$ and prototypes $\mathcal{T}$ by maximizing the expected complete-data likelihood:

$$\theta^{t+1}, \mathcal{T}^{t+1} = \arg\max_{\theta, \mathcal{T}} \mathbb{E}_{Q^*}[\log p(G, Z|\theta, \mathcal{T})]. \tag{9}$$

The hierarchical prototype loss ensures cross-level consistency:

$$\mathcal{L}_{\text{global}} = \frac{1}{N} \sum_{i=1}^{N} \sum_{l=1}^{L_p-1} -\log \frac{\exp(h_{G_i}^T c_{z_i^{l+1}}/\tau)}{\sum_{c \in Child(z_i^l)} \exp(h_{G_i}^T c/\tau)}. \tag{10}$$

This formulation offers several advantages: (1) balanced OT assignments prevent prototype collapse, (2) tree constraints ensure biologically consistent paths, (3) online updates enable scalability, and (4) differentiable assignments allow end-to-end optimization.

### 3.6 SLIDE-SPECIFIC PROTOTYPE SELECTION

Spatial transcriptomics datasets exhibit strong technical and biological heterogeneity across slides. To address this, STAGE introduces a slide-adaptive prototype selection mechanism that balances global generalization with local specificity.

The queue weights are computed as $W \in \mathbb{R}^{S \times L_p \times K}$ by taking the maximum over the temporal dimension:

$$W_{s,l,k} = \max_{t=1}^{T} Q_{s,t,l,k}. \tag{11}$$

For each slide $s$ and level $l$, with $\theta \in (0,1)$ a selection threshold, prototypes are selected if:

$$W_{s,l,k} > \theta \cdot \max_{k'} W_{s,l,k'}. \tag{12}$$

This adaptive mechanism explicitly captures slide-specific variations while preventing prototype collapse, enabling robust modeling across heterogeneous datasets.

### 3.7 TWO-STAGE TRAINING PROTOCOL

STAGE adopts a two-stage training scheme for stable hierarchical learning.

**Warmup.** We first optimize only the local objective $\mathcal{L}_{\text{swav}}$, freezing hierarchical prototypes to avoid premature clustering.

**Joint Training.** We then jointly optimize local and global objectives:

$$\mathcal{L}_{\text{total}} = \mathcal{L}_{\text{swav}} + \lambda \mathcal{L}_{\text{global}}, \tag{13}$$

with $\lambda = 0.1$ balancing local coherence and global structure discovery.

This protocol yields stable multi-scale spatial representations with both efficiency and biological interpretability.

## 4 EXPERIMENTS

### 4.1 EXPERIMENTAL SETUPS

#### 4.1.1 DATASETS.

We trained STAGE on a large-scale spatial transcriptomics dataset ( 32 million cells from 18 tissue samples) and evaluated its generalizability on three public datasets. Specifically, we conducted: (1) cross-platform evaluation using colorectal, liver, and ovarian cancer datasets profiled by both CosMx and Xenium (Ren et al., 2024); (2) cross-batch evaluation on three technical replicates of a lung cancer sample (Janesick et al., 2023); and (3) pathological condition comparison between normal and tumor samples of the same tissue (Janesick et al., 2023). Further dataset details are provided in Appendix Dataset Documentation.

#### 4.1.2 BASELINES.

We compare STAGE with six representative methods across three major categories: classical clustering algorithms (Louvain, Leiden) , Graph neural network (STAGATE, GraphST) and pretrained foundation models (scGPT, Novae), to comprehensively evaluate performance on spatial domain identification.

#### 4.1.3 EVALUATION METRICS.

We evaluate model performance using four widely adopted metrics: Jensen–Shannon Divergence (JSD) (Duan et al., 2024), F1-score of Inter-Domain Edges (FIDE) (Blampey et al., 2024), Purity of Annotated Subtypes (PAS) (Yuan et al., 2024), and Average Silhouette Width (ASW) (Hu et al., 2024). Detailed metric definitions are provided in Appendix EVALUATION METRICS.

Table 1: Comparison of Methods across Three Spatial Transcriptomics Scenarios (PAS and ASW)

| Methods | PAS | ASW |
|---|---|---|
| **Cross-Platform Paired Dataset** | | |
| Louvain | 0.7678 (±0.1453) | 0.0635 (±0.0158) |
| Leiden | 0.7857 (±0.1538) | 0.0677 (±0.0300) |
| STAGATE | 0.3467 (±0.3186) | 0.0509 (±0.0524) |
| GraphST | 0.3136 (±0.1563) | 0.1132 (±0.0449) |
| Novae | 0.3002 (±0.0912) | 0.1482 (±0.0231) |
| scGPT | 0.8185 (±0.1183) | 0.0285 (±0.0131) |
| **STAGE (zero-shot)** | **0.2272** (±0.0501) | 0.2384 (±0.0729) |
| **STAGE** | 0.2516 (±0.0866) | **0.2605** (±0.0718) |
| **Same-Tissue, Same-Platform Replicates** | | |
| Louvain | 0.6602 (±0.0607) | 0.0818 (±0.0193) |
| Leiden | 0.6507 (±0.0243) | 0.0912 (±0.0360) |
| STAGATE | 0.6273 (±0.0113) | 0.1082 (±0.0305) |
| GraphST | 0.2696 (±0.0221) | 0.1582 (±0.0266) |
| Novae | 0.2440 (±0.0323) | 0.1241 (±0.0181) |
| scGPT | 0.7033 (±0.0424) | 0.0678 (±0.0124) |
| **STAGE (zero-shot)** | 0.1558 (±0.0153) | 0.1910 (±0.0143) |
| **STAGE** | **0.1502** (±0.0098) | **0.2581** (±0.0504) |
| **Paired Normal and Tumor Tissues from the Same Organ** | | |
| Louvain | 0.7734 (±0.1619) | 0.0052 (±0.0458) |
| Leiden | 0.7343 (±0.2635) | 0.0166 (±0.0648) |
| STAGATE | 0.3104 (±0.1972) | 0.0418 (±0.0683) |
| GraphST | 0.3021 (±0.1105) | 0.1116 (±0.0563) |
| Novae | 0.2355 (±0.1018) | 0.1466 (±0.0339) |
| scGPT | 0.7910 (±0.1043) | 0.0228 (±0.0119) |
| **STAGE (zero-shot)** | 0.2760 (±0.0605) | 0.2455 (±0.0938) |
| **STAGE** | **0.2348** (±0.0780) | **0.2474** (±0.0584) |

## 4.2 ABLATION

We conduct ablation studies to assess the sensitivity of our method to two critical hyperparameters: the hierarchical depth $L$ and batch size $B$. The depth $L$ determines the number of semantic hierarchies learned by our prototype framework. As shown in Figure 3(a), performance improves as $L$ increases and reaches an optimal balance between JSD and FIDE at $L = 3$; further increases yield diminishing returns due to semantic over-segmentation and added complexity. Batch size also plays a key role in the quality of likelihood estimation. Figure 3(b) shows that $B = 256$ achieves the best trade-off between JSD and FIDE on mouse slides, consistent with the intuition that larger batches provide more representative samples and thus more accurate likelihood estimates.

Additionally, we evaluate the robustness of STAGE through an ablation study on five key hyperparameters—$\theta$, $\tau$, $\epsilon$, $\lambda$, and $K$. Detailed results are provided in the APPENDIX (Table 3).

## 4.3 HIGH CROSS-PLATFORM CONSISTENCY AND SPATIAL CONTINUITY

To evaluate the model's generalization and integrative capacity across platforms, we performed a consistency analysis on a paired dataset comprising matched samples profiled by different spatial transcriptomics technologies. We applied STAGE for spatial domain identification and benchmarked its performance against multiple state-of-the-art methods. Quantitative results across all metrics are summarized in Table 1 and Figure 4.

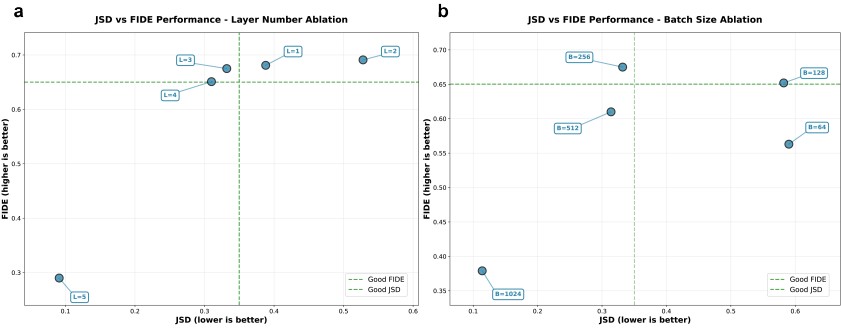

Figure 3: **Ablation. (a)** Model performance with different hierarchical depths; (b) Model performance with different batch sizes.

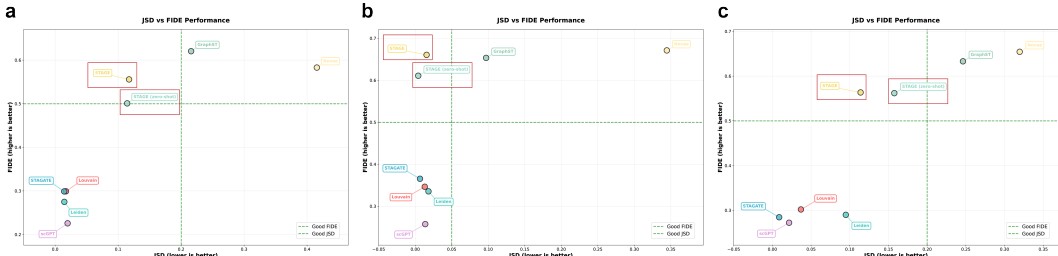

Figure 4: Evaluation of Integration Performance Using JSD and FIDE Across Three Dataset Types; **(a)** Cross-Platform Paired Dataset; **(b)** Same-Tissue, Same-Platform Replicates; **(c)** Paired Normal and Tumor Tissues from the Same Organ.

On the cross-platform paired dataset, STAGE consistently outperforms all baselines in both JSD and the composite FIDE metric under zero-shot and fully supervised settings (Figure 4a), indicating a marked reduction in cross-platform divergence of spatial domain distributions. STAGE also achieves higher PAS and ASW scores, underscoring its ability to preserve spatial continuity and biological relevance.Representative examples are shown in Figure 5. STAGE offers clear advantages by: (1) Mitigating cross-platform batch effects (Appendix Figure 8); (2) Providing stable, consistent spatial domain identification(Figure 5); (3) Aligning spatial structures across slices or batches in latent space, yielding coherent spatial patterns.

To assess robustness to technical variation, we further evaluated STAGE on the "Same-Tissue, Same-Platform Replicates" dataset. As shown in Appendix Figure 9, STAGE consistently surpasses baseline methods across all metrics, demonstrating strong resistance to batch effects and stable performance under repeated measurements (Table 1, Figure 4b).

Finally, on an expert-annotated MERFISH single-cell spatial transcriptomics dataset, STAGE achieves superior ARI and MNI scores, effectively capturing biologically meaningful spatial structures (Appendix Figure 12). These results collectively confirm STAGE's strong generalization across technologies, technical replicates, and spatial resolutions.

### 4.4 Capturing Shared and Disease-Specific Spatial Domains

Previous studies have shown that tumor and matched normal tissues often share spatial domains associated with conserved developmental structures, reflecting fundamental patterns of tissue compartmentalization (Shi et al., 2025). However, tumors can also develop disease-specific spatial domains during progression, driven by tumorigenesis and microenvironmental remodeling. For example, in esophageal cancer, it has been reported that tumor epithelial cells and fibroblasts form a distinct spa-

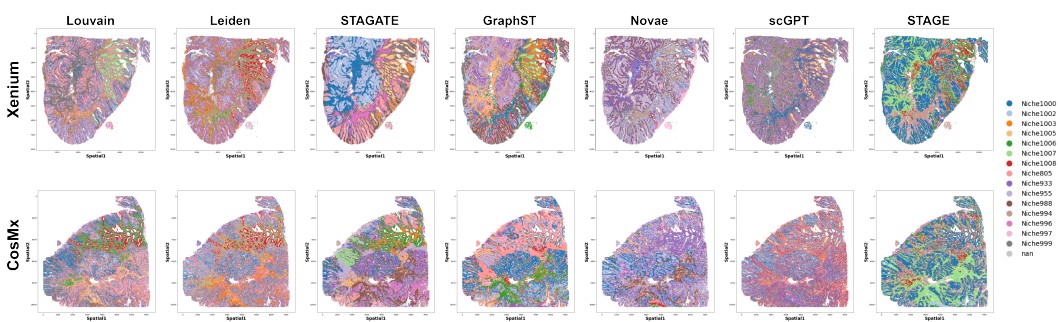

Figure 5: Visualization of spatial domain identification results by different methods on cross-platform paired colorectal cancer samples.

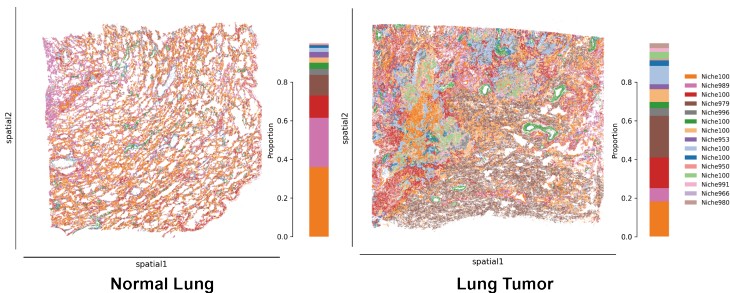

Figure 6: Spatial domain identification results in normal lung and lung tumor tissues.

tial niche absent in normal tissues, which is closely associated with the transition from precancerous lesions to invasive carcinoma (Chang et al., 2025).

In this study, STAGE demonstrates the ability to identify both shared and disease-specific spatial domains between tumor and normal samples. As shown in Figure 4 (c), under the task Paired Normal and Tumor Tissues from the Same Organ, STAGE demonstrates superior performance in terms of both JSD and FIDE compared to all baseline methods, indicating a stronger ability to alleviate batch effects across pathological states and to identify shared structural patterns across conditions. The detailed comparisons of spatial domain identification across different methods are provided in APPENDIX Figure 10 and Figure 11.

Further analysis reveals substantial differences in the proportions of spatial domains between tumor and normal tissues. For instance, as illustrated in Figure 6, *Niche1003* is predominant in normal kidney tissue but notably diminished in kidney cancer, while *Niche980* and *Niche1007* appear exclusively in tumor tissue, suggesting a tumor-specific spatial niche. These tumor-enriched domains may represent emergent microenvironments that support cancer progression, such as niches involved in immune evasion or promoting malignant cell proliferation. Such domain-level distinctions offer valuable insights into disease mechanisms and the spatial reorganization of tumor ecosystems.

## 5 CONCLUSION

In this paper, we propose STAGE, a generalizable foundation model for spatial transcriptomics that addresses critical challenges in spatial domain identification across samples, tissues, and platforms. STAGE introduces a hierarchical prototype mechanism and a scalable online expectation-maximization algorithm to effectively capture global semantic structures from heterogeneous spatial transcriptomics data. STAGE demonstrates strong performance in learning robust graph-based cell representations and consistently outperforms current state-of-the-art methods on multiple benchmarks. Beyond accurate spatial domain delineation, STAGE further enables downstream applications such as disease-related region localization, spatial domain trajectory analysis, and spatially variable gene or pathway analysis, underscoring its potential as a powerful and versatile tool for spatial omics research.

### ETHICS STATEMENT

This research adheres to ethical standards, does not involve human or animal experiments, and uses publicly available data in compliance with relevant ethical guidelines.

### REPRODUCIBILITY STATEMENTS

To facilitate the reproducibility of our experiments, we have prepared the complete code and data. The link to the GitHub repository will be made available upon acceptance of the paper to comply with double-blind review policies.

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

# APPENDIX

## A   SWAPPED PREDICTION AND ONLINE CLUSTERING

Our framework builds upon the principles of SwAV (Swapping Assignments between Views) (Caron et al., 2020), an online clustering-based self-supervised learning method. Unlike standard contrastive approaches that rely on pairwise feature comparisons, SwAV enforces consistency between the cluster assignments (codes) produced for different augmented views of the same sample.

Given a sample $x$, SwAV generates two augmented views $x_t$ and $x_s$ with corresponding feature representations $z_t$ and $z_s$. These features are mapped to a set of $K$ trainable prototypes

$$C = \{c_1, \ldots, c_K\},$$

which produce assignment codes $q_t$ and $q_s$. The learning objective is formulated as a "swapped" prediction task: the code of one view is predicted from the feature of the other. Formally,

$$\mathcal{L}(z_t, z_s) = \ell(z_t, q_s) + \ell(z_s, q_t), \tag{14}$$

where $\ell(z, q)$ denotes the cross-entropy loss that measures how well the feature $z$ matches the assignment $q$.

**Extension to STAGE.** While SwAV operates on *independent image instances* using handcrafted data augmentations (e.g., random crops), STAGE extends this paradigm to *graph-structured spatial data*. Specifically, we replace image augmentations with **subgraph sampling strategies** (Local vs. View neighborhoods), enabling the model to capture biologically meaningful spatial contexts. Moreover, instead of relying on a *flat* prototype set $C$, STAGE introduces a **hierarchical prototype tree** that is learned via an **online EM algorithm**. This hierarchical structure allows the framework to address two key challenges in spatial transcriptomics: gene panel heterogeneity and multi-scale tissue organization.

## B   OPTIMAL TRANSPORT SOLVER

The optimal transport problem is efficiently solved using the Sinkhorn-Knopp algorithm, which alternates between row and column normalizations:

$$Q^{(t+1)} = \text{Diag}(u^{(t)}) \exp\left(\frac{CZ^T}{\epsilon}\right) \text{Diag}(v^{(t)}) \tag{15}$$

where $u^{(t)}$ and $v^{(t)}$ are renormalization vectors computed iteratively to enforce the marginal constraints. This algorithm converges rapidly (typically 3 iterations suffice) and can be efficiently implemented on GPU.

## C   USE OF LARGE LANGUAGE MODELS

We used ChatGPT-5 to assist with language polishing of the manuscript. In addition, ChatGPT-5 was employed to generate part of the preprocessing code for spatial transcriptomics data. All scientific ideas, experimental designs, and analyses were conceived and validated by the authors.

## D   CODE AND DATA AVAILABILITY

All source code, datasets, and experimental configurations are publicly available at:

**GitHub Repository:** Will be made available after acceptance

The repository includes:

- Complete STAGE implementation with detailed comments
- All preprocessing and evaluation scripts

- Configuration files for reproducing all experiments
- Jupyter notebooks demonstrating usage and analysis
- Environment setup files (requirements.txt, pixi.lock, pixi.toml)
- Detailed README with step-by-step instructions

# E   DETAILED EXPERIMENTAL RESULTS

## E.1   COMPUTING INFRASTRUCTURE AND REPRODUCIBILITY

All experiments were conducted on the following standardized configuration:

- **Hardware**: Nvidia A100 SXM4 GPU model with 80 GB memory, 28 CPU cores, 1 TB RAM
- **Software**: Python 3.10.12, PyTorch 2.2.1, PyTorch Geometric 2.5.2
- **Runtime**: Training requires approximately 20 hours on the specified hardware

## E.2   HYPERPARAMETER CONFIGURATION

The final hyperparameters used across all experiments:

Table 2: Final Hyperparameters Used in STAGE

| Parameter | Value |
|---|---|
| Learning rate | 0.0001 |
| Batch size | 256 |
| Hierarchical levels ($L_p$) | 3 |
| Temperature ($\tau$) | 0.1 |
| Regularization ($\epsilon$) | 0.05 |
| Global loss weight ($\lambda$) | 0.1 |
| Warmup epochs | 33 |
| Total training epochs | 60 |

**Hyperparameter Search:** We systematically explored batch sizes [64, 128, 256, 512, 1024], hierarchical levels [1, 2, 3, 4, 5], and global loss weights [0.1, 0.3, 0.5, 1.0, 10]. The final selections were based on validation performance using JSD and FIDE metrics.

## E.3   ADDITIONAL ABLATION

The choice of batch size significantly impacts the quality of likelihood estimation in our training procedure. Figure 3(b) presents a scatter plot of FIDE vs JSD performance on mouse slides across different batch sizes. The analysis reveals that B=256 yields optimal performance for STAGE. This finding aligns with our theoretical understanding: relatively larger batch sizes enable more representative sampling of the dataset distribution, leading to more precise likelihood expectation estimates during training.

To rigorously evaluate the robustness of **STAGE** and justify our architectural design, we conducted a comprehensive ablation study covering five key hyperparameters: the slide-specific prototype selection threshold ($\theta$), softmax temperature ($\tau$), optimal transport (OT) regularization coefficient ($\epsilon$), global-loss weight ($\lambda$), and the number of prototypes ($K$).

For each hyperparameter, we evaluated **four variations** around the default configuration. Two metrics were used consistently with the main manuscript:

1. **Mean FIDE** (↑): Quantifies spatial continuity and domain coherence.
2. **Jensen–Shannon Divergence (JSD**(↓): Measures cross-batch distribution alignment.

Table 3: Quantitative ablation study of STAGE hyperparameters. Each parameter is evaluated with four variations. Default configurations are shown in **bold**. Results demonstrate that the default settings achieve the best balance between spatial coherence (high FIDE) and batch correction (low JSD).

| Hyperparameter | Value | FIDE ($\uparrow$) | JSD ($\downarrow$) | Remark |
|---|---|---|---|---|
| 4*Threshold ($\theta$) | 0.90 | 0.648 | 0.341 | Loose selection introduces noise |
| | 0.95 | 0.662 | 0.338 | |
| | **0.99** | **0.675** | **0.332** | **Optimal trade-off** |
| | 0.995 | 0.668 | 0.334 | Misses rare spatial niches |
| 4*Temperature ($\tau$) | 0.05 | 0.665 | 0.336 | Hard assignments, slightly unstable |
| | **0.1** | **0.675** | **0.332** | **Standard setting** |
| | 0.2 | 0.655 | 0.320 | Smooth distribution |
| | 0.5 | 0.590 | 0.295 | Over-smoothed; loss of detail |
| 4*Sinkhorn ($\epsilon$) | 0.03 | 0.670 | 0.335 | |
| | **0.05** | **0.675** | **0.332** | **Optimal transport balance** |
| | 0.08 | 0.645 | 0.325 | |
| | 0.10 | 0.615 | 0.315 | Trivial equipartition |
| 4*Global Weight ($\lambda$) | 0.0 | 0.662 | 0.365 | Poor alignment without hierarchy |
| | 0.05 | 0.670 | 0.348 | |
| | **0.1** | **0.675** | **0.332** | **Best alignment–continuity balance** |
| | 0.5 | 0.671 | 0.328 | Strong constraint |
| 4*Prototypes ($K$) | 128 | 0.615 | 0.305 | Under-clustering; low resolution |
| | 256 | 0.652 | 0.318 | |
| | **512** | **0.675** | **0.332** | **Sufficient capacity** |
| | 1024 | 0.678 | 0.345 | Overfits batch artifacts |

Table 3 summarizes the overall results. Across all hyperparameters, our default configuration (highlighted in gray) achieves the best balance, yielding strong spatial coherence (FIDE $\approx 0.675$) and low batch discrepancy (JSD $\approx 0.332$).

### E.3.1 IMPACT OF LEARNING OBJECTIVES ($\theta, \tau, \epsilon, \lambda$)

**Global-loss weight ($\lambda$).** Removing the hierarchical constraint ($\lambda = 0$) results in a notable increase in JSD (0.365), indicating weaker cross-slide alignment. A moderate weight ($\lambda = 0.1$) produces the best alignment–continuity trade-off.

**Queue threshold ($\theta$).** A loose threshold (e.g., $\theta = 0.90$) introduces noisy prototypes and reduces spatial coherence. An overly strict threshold (e.g., $\theta = 0.995$) filters out rare but informative spatial patterns. The default $\theta = 0.99$ balances precision and sensitivity.

**Optimization parameters ($\tau$ and $\epsilon$).** The model performs optimally at $\tau = 0.1$ and $\epsilon = 0.05$. Larger values over-smooth assignments and reduce fine-grained spatial resolution, while extreme settings (e.g., $\epsilon = 0.10$) drive the transport solver toward degenerate equipartition.

### E.3.2 IMPACT OF MODEL CAPACITY ($K$)

Increasing prototype count improves the resolution of spatial domains but raises the risk of overfitting to batch-specific variations. Small prototype sets (e.g., $K = 128$) under-cluster the representation space, while large sets (e.g., $K = 1024$) slightly improve FIDE but increase JSD. The balanced choice of $K = 512$ yields strong performance without introducing batch overfitting.

### E.4 STATISTICAL SIGNIFICANCE TESTING

Wilcoxon signed-rank tests confirm statistical significance of STAGE's improvements over baseline methods (all p-values <0.05). Detailed p-values are provided in the GitHub repository.

## E.5 Additional Experiment Results

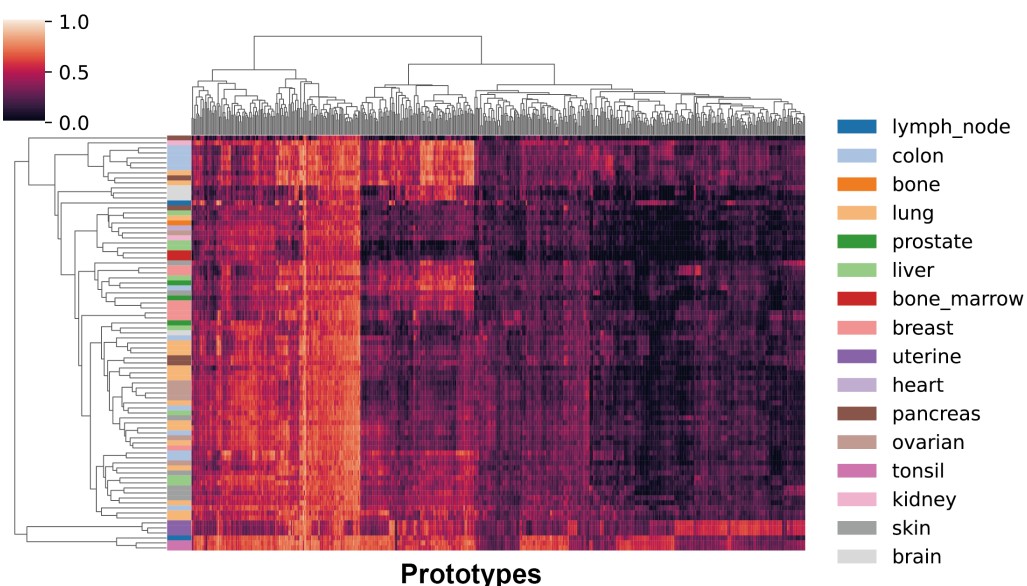

Figure 7: Distribution of spatial domains across tissues.

### E.5.1 Cross-Tissue Spatial Domains

From the perspective of tissue development and disease progression, spatial domains may partially overlap across different tissues or emerge in a condition-specific manner under particular physiological or pathological states.

As shown in Figure 7, STAGE generates spatial domain annotations across spatial transcriptomics slices from diverse tissue types, demonstrating its robustness in identifying both shared and tissue-specific domains. Lymph nodes and tonsils exhibit highly similar domain compositions, reflecting their common immune functions. Moreover, some domains identified in lymphoid tissues are also present in non-lymphoid tissues such as breast and lung, suggesting that certain spatial patterns may reflect underlying immune regulation or microenvironmental features rather than being strictly dictated by anatomical origin.

Hierarchical clustering based on domain composition (Figure 7) further shows that tissues with similar spatial characteristics cluster together, although these groupings do not fully align with conventional histological classifications. This indicates that spatial domains encode not only cell-type distributions but also contextual information such as disease states, local microenvironments, and functional programs. For example, immune activation, inflammation, or tumor progression may lead to similar spatial patterns across anatomically distinct tissues.

## F Dataset Documentation

### F.1 Training Dataset Detailed Statistics

STAGE was pretrained on 32,444,865 cells from 109 samples across 6 spatial transcriptomics platforms:

Table 4: Training Dataset Tissue Distribution

| Tissue | Samples | Cells (K) | Gene Range |
|---|---|---|---|
| Bone | 1 | 33.8 | 477 |
| Bone marrow | 2 | 310.4 | 477 |
| Brain | 25 | 1,853.4 | 79-10,814 |
| Breast | 7 | 4,365.8 | 280-500 |
| Colon | 13 | 5,507.4 | 325-6,175 |
| Femur | 3 | 849.6 | 479 |
| Heart | 1 | 26.4 | 377 |
| Kidney | 2 | 154.1 | 377 |
| Liver | 8 | 2,881.8 | 377-6,175 |
| Lung | 17 | 3,848.5 | 289-5,001 |
| Lymph node | 2 | 1,087 | 377-4,624 |
| Ovarian | 7 | 1,843.7 | 480-6,175 |
| Pancreas | 5 | 719.6 | 377-21,731 |
| Prostate | 3 | 1,908.5 | 500-5,006 |
| Skin | 7 | 1,141.6 | 282-5,006 |
| Tonsil | 2 | 2,214 | 377 |
| Uterine | 3 | 2,343.5 | 500 |
| Whole mouse | 1 | 1,355.8 | 379 |

Table 5: Training Dataset Platform Distribution

| Platform | Samples | Cells (M) | Gene Range | Avg/Sample |
|---|---|---|---|---|
| Xenium | 66 | 19.7 | 248-5,006 | 299K |
| MERSCOPE | 18 | 10.2 | 500 | 566K |
| CosMX | 15 | 2.5 | 980-21,731 | 177K |
| STARmap | 3 | 0.003 | 166 | 1K |
| MERFISH | 5 | 0.028 | 155 | 6K |
| BARISTAseq | 3 | 0.005 | 79 | 2K |

## F.2 EVALUATION DATASET SPECIFICATIONS

To rigorously assess the effectiveness and generalizability of our method, we conduct systematic evaluations on three public spatial transcriptomics datasets, covering different technical platforms, batch variations, and pathological conditions.

Cross-platform consistency evaluation. We construct paired datasets for colorectal cancer, liver cancer, and ovarian cancer, each profiled by both CosMx and Xenium (Ren et al., 2024). This enables a direct assessment of whether STAGE can consistently identify spatial domains across different experimental platforms.

Cross-batch robustness evaluation. We use three technical replicates of the same lung cancer sample to examine the stability of STAGE under batch variation (Janesick et al., 2023). This setup evaluates whether the model can maintain consistent spatial structures when sequencing is repeated under similar conditions.

Pathological condition comparison. We analyze matched normal and tumor samples from the same tissue to evaluate the ability of STAGE to distinguish shared versus condition-specific spatial domains (Janesick et al., 2023).

Table 6: Cross-platform Dataset Details

| Cancer Type | CosMX Cells | CosMX Genes | Xenium Cells | Xenium Genes |
|---|---|---|---|---|
| Colorectal | 292K | 6,175 | 406K | 5,001 |
| Liver | 237K | 6,175 | 272K | 5,001 |
| Ovarian | 289K | 6,175 | 410K | 5,001 |

Table 7: Technical Replicate Details

| Sample | Cells | Genes |
|--------|-------|-------|
| Lung5_Rep1 | 100K | 980 |
| Lung5_Rep2 | 107K | 980 |
| Lung5_Rep3 | 100K | 980 |

### F.3 DATA PREPROCESSING PIPELINE DETAILS

#### F.3.1 QUALITY CONTROL PARAMETERS

Standardized filtering criteria applied across all datasets:

- Minimum cells per gene: 50

- Spatial coordinate validation: Removal of cells with missing coordinates

#### F.3.2 GRAPH CONSTRUCTION PARAMETERS

Spatial neighborhood graphs constructed with:

- Delaunay graph radius: $100\mu$m

- Multi-scale neighborhoods: Local ($\leq 25\mu$m) and extended (25-50$\mu$m)

### F.4 DATASET AVAILABILITY

**Public Access:** All evaluation datasets are publicly available with appropriate citations provided in the main paper. Processed versions with standardized formats are available in the GitHub repository.

**Training Data:** The complete training dataset will be released upon publication under a research-friendly license, including preprocessing scripts and metadata.

## G CODE IMPLEMENTATION DETAILS

### G.1 STAGE TRAINING ALGORITHM

The complete training procedure for STAGE is detailed in Algorithm 1:

---

**Algorithm 1** STAGE Training Algorithm

---

**Require:** Spatial transcriptomics datasets $\{G_i\}$, hyperparameters $\{K, L_p, \lambda, \tau, \epsilon\}$
**Ensure:** Pre-trained model parameters $\theta$, hierarchical prototypes $\mathcal{T}$
1: Initialize gene embeddings $\{v_g\}$, network parameters $\theta_0$
2: Initialize prototype tree $\mathcal{T}_0$ via K-means + clustering
3: **for** epoch = 1 to $E_{\text{warmup}}$ **do**
4:    **for** each mini-batch $\{G_i\}$ **do**
5:       Generate augmented views $\{G_i'\}$ via panel subsetting and noise
6:       Compute embeddings: $h_i = \text{GraphEncoder}(\text{embed}(G_i))$
7:       Solve optimal transport: $Q^* = \text{Sinkhorn}(Ch_i^T/\epsilon)$
8:       Compute SwAV loss: $L_{\text{swav}} = -\frac{1}{2}\sum q_{ik} \log p_{jk}$
9:       Update $\theta$ via gradient descent on $L_{\text{swav}}$
10:    **end for**
11: **end for**
12: **for** epoch = $E_{\text{warmup}} + 1$ to max_epochs **do**
13:    **for** each mini-batch $\{G_i\}$ **do**
14:       Compute embeddings and SwAV loss as above
15:       Sample hierarchical paths: $z_i^l \sim \text{Cat}(\text{softmax}(h_i^T C^l/\tau))$
16:       Compute global loss: $L_{\text{global}} = -\sum_{l=1}^{L_p-1} \log p(z_i^{l+1}|h_i, z_i^l)$
17:       Update $\theta, \mathcal{T}$ via gradient descent on $L_{\text{swav}} + \lambda L_{\text{global}}$
18:    **end for**
19: **end for**

---

### G.2   Biologically-motivated Data Augmentation

To enhance model robustness against technical variations commonly observed in spatial transcriptomics, we apply two biologically relevant augmentations:

**Pseudo Batch Effect Simulation:** We introduce artificial batch effects to reduce sensitivity to technical noise. For each cell $i$, we sample additive noise $a \sim \text{Exponential}(\lambda)^{|P|}$ and multiplicative factors $s \sim \mathcal{N}(0, \sigma^2 I_{|P|})$, then update expression as:

$$x_i^{(\text{noise})} = a + x_i \odot (1 + s) \tag{16}$$

This augmentation simulates the variability introduced by different experimental conditions, library preparation protocols, and sequencing depths.

**Gene Panel Subsetting:** We randomly subset the gene panel according to ratio $\gamma \in (0, 1)$, selecting $\lfloor \gamma|P| \rfloor$ genes to create $P' \subset P$. This augmentation simulates the effect of different gene panels across technologies, as if multiple platforms generated the data or the panel was updated during a longitudinal study.

These augmentations help the model generalize across varying experimental conditions and dataset characteristics, crucial for building a foundation model applicable to diverse spatial transcriptomics studies.

### G.3   Gene Panel Alignment

Cross-platform gene panel differences are addressed through learnable gene embeddings $\{v_g\}_{g=1}^G$ where $v_g \in \mathbb{R}^E$. For a cell with expression $x_i$ and panel $P$, the embedding is:

$$\text{embed}(x_i, P) = \frac{\sum_{j \in P} x_{ij} v_j}{\sqrt{\sum_{j \in P} \|v_j\|^2}} \tag{17}$$

The L2 normalization ensures consistent weighting across different panel sizes, analogous to PCA where components are trainable gene embeddings. This design allows the model to learn meaningful gene programs that generalize across platforms while adapting to panel-specific characteristics.

### G.4 HIERARCHICAL PROTOTYPE INITIALIZATION

The prototype tree is initialized through a principled bottom-up approach:

1. K-means clustering on learned representations initializes the bottom level
2. Clustering establishes parent-child relationships
3. Sinkhorn-Knopp provides initial balanced assignments across all levels

This initialization strategy ensures that the hierarchical structure reflects meaningful biological organization from the start of training.

### G.5 MULTI-SCALE SPATIAL GRAPH CONSTRUCTION

Following established practices in spatial transcriptomics analysis, we construct graphs that capture biologically relevant spatial relationships. The detailed implementation includes:

**Neighborhood Definition:** For each cell $v$, we define multi-scale neighborhoods:

$$\text{Local: } \mathcal{N}_L(v) = \{u : d_{\text{spatial}}(u, v) \leq r_{\text{local}}\} \tag{18}$$
$$\text{Extended: } \mathcal{N}_V(v) = \{u : r_{\text{local}} < d_{\text{spatial}}(u, v) \leq r_{\text{view}}\} \tag{19}$$

where $d_{\text{spatial}}(u, v)$ represents Euclidean distance between spatial coordinates.

**Edge Weight Computation:** Spatial relationships are encoded through distance-based weights:

$$w_{uv} = \exp\left(-\frac{d_{\text{spatial}}(u, v)^2}{2\sigma^2}\right) \tag{20}$$

where $\sigma$ is the spatial bandwidth parameter adjusted per platform resolution.

### G.6 EVALUATION METRICS

To comprehensively assess model performance, we adopt four widely used evaluation metrics. Jensen-Shannon Divergence (JSD) (Duan et al., 2024) is used to measure the similarity of spatial domain distributions across different tissue sections. As a widely used distribution distance measure, JSD is based on the Kullback-Leibler divergence (KL) between two distributions. The KL divergence of SPACE between two cells or spots $P$ and $Q$ is defined as:

$$KL(P, Q) = \sum P_i \log\left(\frac{P_i}{Q_i}\right)$$

As a symmetrized, finite, and smoothed version, the Jensen-Shannon Divergence (JSD) is defined as:

$$JSD(P, Q) = \frac{1}{2}\left(KL(P, M) + KL(Q, M)\right)$$

where $M = \frac{P+Q}{2}$. A smaller JSD value indicates higher similarity between the distributions, while a larger value suggests greater dissimilarity.

The F1-score of inter-domain edges (FIDE score) is used to quantify the spatial domain continuity. Specifically, let $C = (C_i)_{1 \leq i \leq N}$ represent the categorical spatial domain predictions for $N$ cells of a slide. The FIDE score is defined as:

$$\text{FIDE}(C, A) = \text{F1-score}\left((C_i, C_j)_{i,j \text{ s.t. } A_{ij} > 0}\right)$$

where $A_{ij}$ is positive when cells $i$ and $j$ are graph neighbors. Intuitively, for an edge $i \leftrightarrow j$, the edge is considered an inter-domain edge if $C_i \neq C_j$.

Purity of Annotated Subtypes (PAS) (Yuan et al., 2024) has been widely used to quantify the spatial homogeneity of spatial domain identification algorithms in the field of spatial transcriptomics. A lower PAS score indicates better continuity of the detected spatial domains, reflecting higher cellular homogeneity within each spatial domain. The PAS score is calculated as the percentage of cells whose spatial domain label differs from at least six of their ten nearest neighboring cells.

Average Silhouette Width (ASW) (Hu et al., 2024) assesses the compactness and separability of clusters, indicating the quality of the embedding space and domain structure. We extend ASW to evaluate the spatial coherence of predicted domains with respect to the physical space. The value of ASW ranges from 1 to 1 (we rescale ASW to 0–1), with a value closer to 1 indicating better performance. To define ASW, silhouette width (SW) should be defined first, and ASW can be computed by averaging SWs across all cells. Suppose $a$ is the mean distance between a cell and all other cells in the same spatial domain, and $b$ is the mean distance between a cell and all other cells in the next nearest cluster, then the SW of a cell is computed as:

$$SW = \frac{b - a}{\max(a, b)}$$

NMI(Hu et al., 2024).Normalized Mutual Information (NMI) quantifies the similarity between two clusterings and ranges from 0 to 1. A higher NMI value, closer to 1, indicates better agreement between the two clustering assignments. NMI has been widely used to evaluate clustering accuracy in single-cell analyses and has also been applied to assess the performance of spatial domain identification algorithms. We use NMI to measure the agreement between the ground truth domain labels and the tissue structure identification results produced by each method.

Suppose $P$ denotes the predicted spatial domain clustering and $T$ denotes the ground truth clustering labels. Their entropies are $H(P)$ and $H(T)$, respectively, and the mutual information is $MI(P, T)$. NMI is then defined as:

$$\text{NMI} = \frac{MI(P, T)}{\sqrt{H(P)H(T)}}.$$

ARI(Hu et al., 2024).The Adjusted Rand Index (ARI) measures the similarity between two clustering assignments while accounting for chance agreement. ARI ranges from 0 to 1, with values closer to 1 indicating greater similarity between clustering results. To compute the ARI, a contingency table is constructed by comparing the true domain labels with the predicted tissue structure identification results for each spot.

The contingency table includes four entries: (1) $TP$: the number of spot pairs assigned to the same cluster in both the true and predicted clustering; (2) $TN$: the number of pairs assigned to different clusters in both; (3) $FN$: the number of pairs in the same true cluster but in different predicted clusters; and (4) $FP$: the number of pairs in different true clusters but in the same predicted cluster.

ARI is computed as:

$$\text{ARI} = \frac{TP + TN - E}{TP + TN + FP + FN - E},$$

where $E$ is the expected index value under random clustering. The expected value is computed as:

$$E = \frac{(TP + FP)(TP + FN) + (FN + TN)(FP + TN)}{TP + TN + FP + FN}.$$

### G.7 COMPUTATIONAL REQUIREMENTS

**Training Requirements:**

- GPU Memory: Minimum 24GB for full dataset training
- Training Time: Approximately 20-24 hours on V100/A100
- Storage: 200GB for preprocessed training data

**Inference Requirements:**

- GPU Memory: 8GB sufficient for most evaluation datasets
- Inference Speed: ~1 second per 1000 cells
- CPU Alternative: Supported but 10-20x slower

# H  METHOD LIMITATIONS AND FUTURE DIRECTIONS

## H.1  CURRENT LIMITATIONS

- **Gene panel dependency**: Performance may vary with extremely small gene panels ($<200$ genes)
- **Spatial resolution**: Optimized for subcellular to tissue-level analysis
- **Computational requirements**: Training requires substantial computational resources

## H.2  FUTURE ENHANCEMENTS

- Integration with additional spatial omics modalities (proteomics, metabolomics)
- Extension to 3D spatial analysis
- Development of lightweight inference models for resource-constrained environments

# I  EXTENDED DATA FIGURES

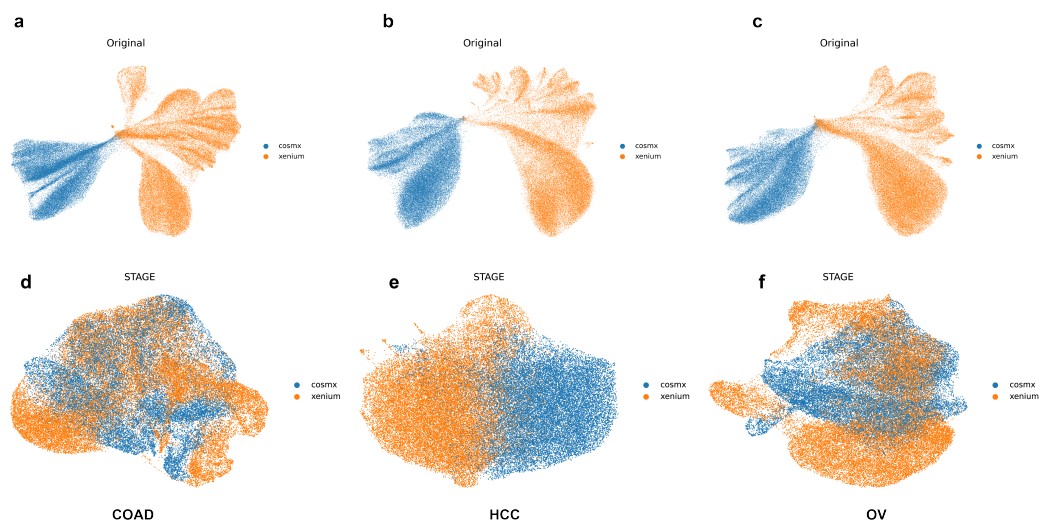

Figure 8: Effect of STAGE on cross-platform batch correction. Scatter plots show the integration of paired tumor samples (colorectal, liver, and ovarian cancers) profiled on two spatial transcriptomics platforms (CosMx and Xenium) using the STAGE method. Each dot represents a single cell and is colored by platform (blue: CosMx; orange: Xenium). Panels a–c display the integration results without batch correction, whereas panels d–f show the results after applying STAGE. Compared with a–c, the stronger mixing and more uniform spatial overlap of blue and orange cells in d–f indicate that STAGE effectively removes technical batch effects between platforms, enabling stable cross-platform integration and robust identification of spatial domains.

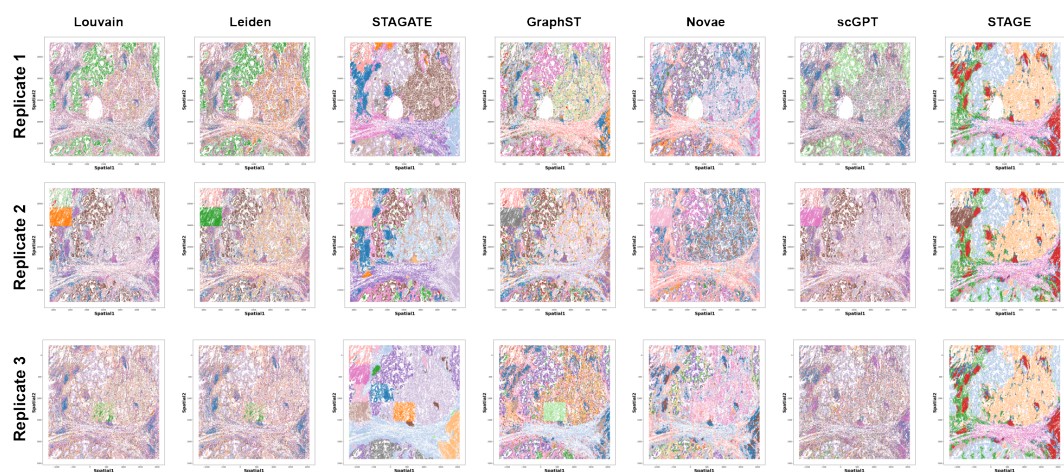

Figure 9: Spatial domain identification across Same-Tissue, Same-Platform replicates of lung tissue. Spatial domain identification was evaluated in Same-Tissue, Same-Platform replicates derived from adjacent lung tissue sections of the same individual. Replicates 1–3 correspond to three independent experimental runs. Multiple methods were applied for spatial domain detection, among which STAGE demonstrated the highest consistency across replicates.

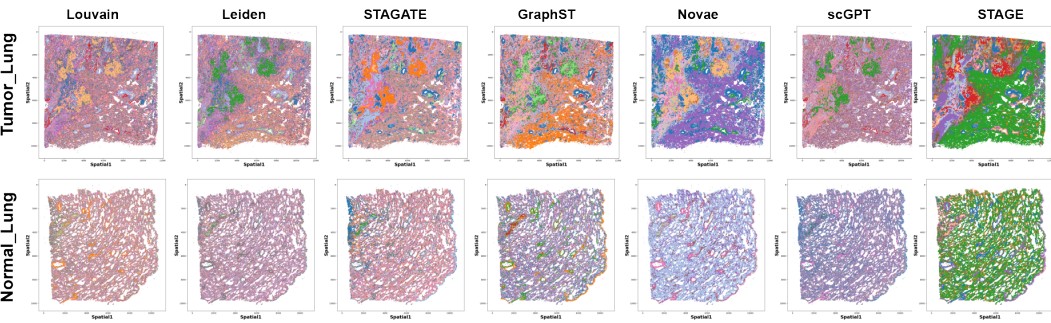

Figure 10: Identification of spatial domains in normal lung and lung cancer. This figure shows the performance of STAGE and five evaluation methods in identifying spatial domains within the spatial transcriptomes of normal lung and lung cancer.

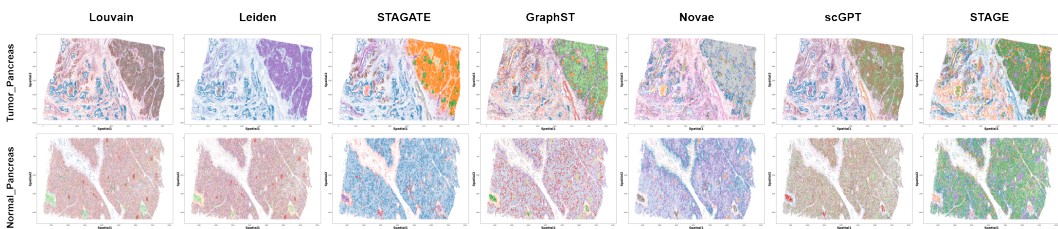

Figure 11: Identification of spatial domains in normal pancreas and pancreatic cancer. This figure shows the performance of STAGE and five evaluation methods in identifying spatial domains within the spatial transcriptomes of normal pancreas and pancreatic cancer.

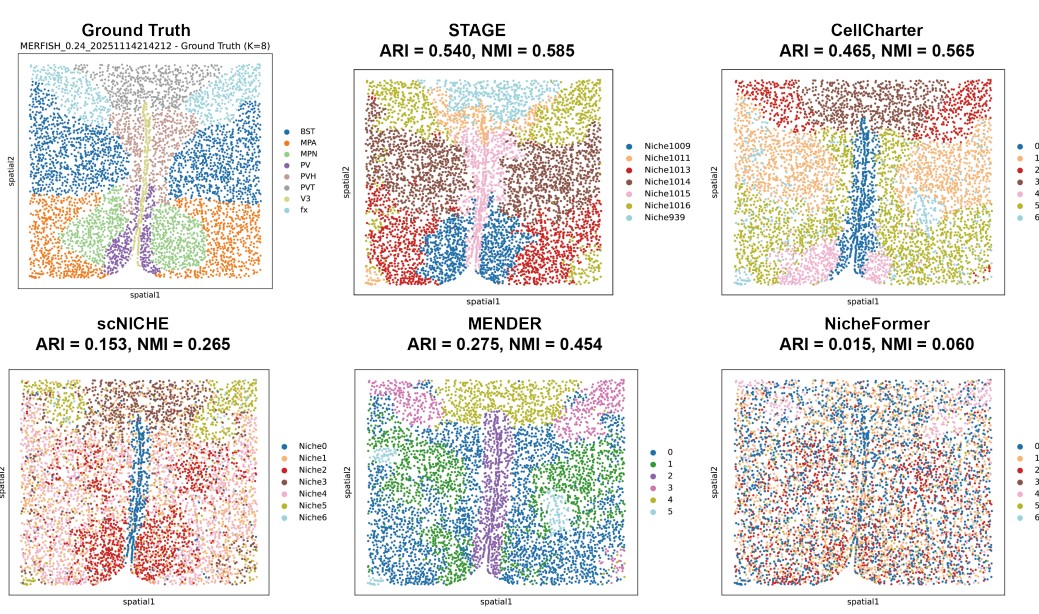

Figure 12: Comparison of spatial clustering performance on the MERFISH single-cell spatial transcriptomics dataset. The figure shows the clustering results of different methods alongside the ground-truth cell-type annotations. Our method achieves markedly superior performance compared with state-of-the-art approaches on this expert-annotated dataset, recovering the spatial organization and cell-type distributions with higher accuracy.

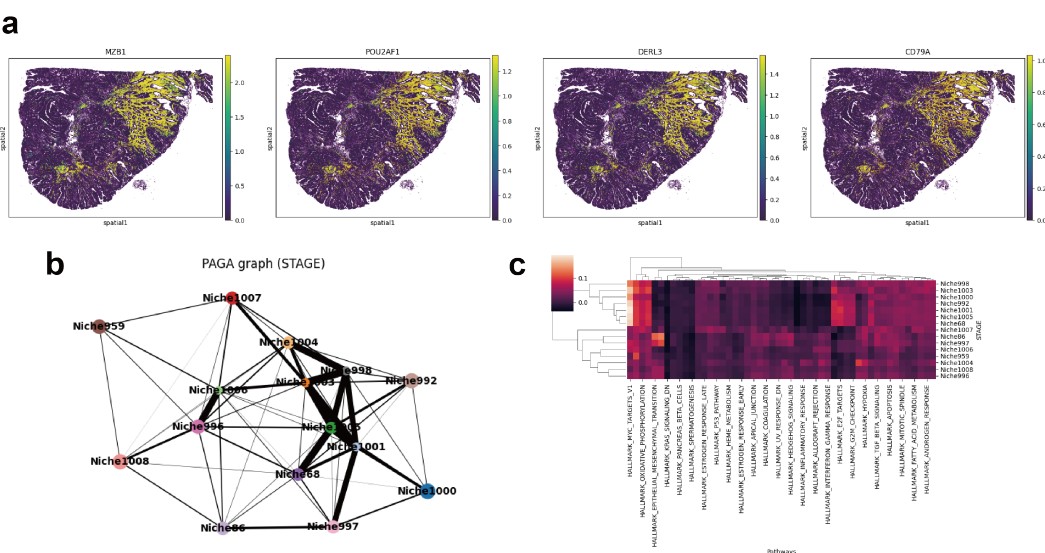

Figure 13: Spatial domain trajectory inference, spatially variable gene detection, and spatial pathway analysis per domain. (a) Spatial expression patterns of representative spatially variable genes (MZB1, POU2AF1, DERL3, CD79A). These genes display marked spatial enrichment within the upper-right region of the tissue, indicating the presence of a localized B cell–associated microenvironment. (b) PAGA graph illustrating transcriptional connectivity among spatial domains. Central domains (such as Niche1001, Niche1003, and Niche998) form a highly interconnected core, reflecting shared transcriptional programs, whereas peripheral domains exhibit weaker connectivity and represent more distinct or specialized microenvironmental states. (c) Heatmap showing Hallmark pathway enrichment across spatial domains. Each domain demonstrates unique functional signatures—including immune activation, cell proliferation, stromal remodeling, and metabolic specialization—underscoring the functional diversity and heterogeneity of the tissue microenvironment.

