# OpenReview forum: "STAGE: A Foundation Model for Spatial Transcriptomics Analysis via Graph Embeddings with Hierarchical Prototypes"
_ICLR.cc/2026/Conference — Submitted to ICLR 2026_

### Official Review · Reviewer_24EE · 2025-10-23

**Soundness:** 3
**Presentation:** 3
**Contribution:** 1
**Rating:** 2
**Confidence:** 5

**Summary:**

The paper introduces a self-supervised framework for learning tissue zonation patterns from spatial transcriptomics data. Drawing inspiration from the classical SwAV framework [2020], the authors propose a clustering-based approach in which a prototype network is trained to align multiple augmented views of the same data point to a shared set of prototypes. Unlike SwAV, however, the presented method extends this principle to the spatial-omics domain and incorporates a hierarchical prototype structure, allowing the model to capture both global tissue organization and finer-grained local zones.

**Strengths:**

The hierarchical prototype assignment is arguably the most compelling aspect of the paper. However, similar extensions of the SwAV framework have already been explored in the computer vision domain \cite{guo_hcsc_2022, xu_hirl_2022}, and related ideas of multiscale niche definition have also been introduced in spatial transcriptomics, for example in MENDER \cite{yuan_mender_2024}. Therefore, while the implementation presented here is technically interesting, it cannot be regarded as a fundamentally novel contribution. Nonetheless, I would encourage the authors to further analyze and interpret the resulting hierarchies, as they may reveal biologically meaningful patterns or inspire new downstream applications.

Authors then benchmark the method against general clustering algorithms (louvain, leiden), cell reprsentation models (scGPT) and a previous generation of niche identification methods (STAGATE [2022], GraphST [2023], Novae [2024]). The focus of benchmarking is primarily on the geometrical properties of the idetnified regions (continuity as measured by FIDE, PAS and ASW scores) and slide integration (estimated using JSD). The paper tries to emphasize that the approach is able to integrate between the tissue slices of (i) one tissue, (ii) multiple tissues and (iii) across the experimental platforms (CosMX and Xenium). The latter is achieved through scGPT-inspired representation of gene expression as a sequence of gene tokens. Though the benchmarking of the approaches for niche identification is not trivial due to the absence of ground truth data in most cases, the paper would benefit from extending (A) the range of the metrics and (B) the suite of the methods tested.

**Weaknesses:**

Numerous existing frameworks already address domain or niche identification, ranging from graph neural networks to simpler yet effective static feature aggregation methods. Moreover, the proposed mechanism for tissue integration is not original to this work but directly borrowed from scGPT; in principle, scGPT-derived embeddings could be integrated into any existing pipeline, the same effect can also be reached with cell type based approach.

The paper’s effort to integrate datasets across tissues and assay platforms is commendable, particularly given the limited development of multitissue models—apart from frameworks such as NicheFormer \cite{schaar_nicheformer_2024}. Nevertheless, the presented results do not convincingly demonstrate the success of this integration. The UMAP visualizations (Figure 8) reveal little to no alignment between CosMX and Xenium datasets, casting doubt on whether joint training across these sources yields any tangible benefit. Moreover, in quantitative comparisons, STAGATE achieves substantially better integration (as reflected by lower JSD values) while employing a considerably simpler modeling approach.

It should also be noted that, from a biological standpoint, cross-tissue integration is a nuanced objective. While certain structural elements recur across organs—such as tertiary lymphoid structures or components of the connective tissue—many aspects of niche organization remain inherently tissue-specific and should not be  overintegrated. I therefore recommend that the authors adjust their benchmarking procedure to account for this nuance. A possible step in this direction could involve comparing slices originating from the same tissue (but critically different donors) versus those from distinct tissues, showcasing that the tissue specific differences are well-preserved.

**Questions:**

Below I suggest modifications that can help improve the paper.

1. **Extend the benchmarking.**
   - Acknowledge that **Louvain**, **Leiden**, and **scGPT** are *not* niche-identification methods; because they ignore explicit cell neighborhoods, they naturally recover *cell identity* rather than *niche identity*. While **STAGATE (2022)**, **GraphST (2023)**, and **Novae (2024)** are genuine niche-discovery algorithms, they are now outdated. Please compare against more recent methods such as *MENDER*, *CellCharter*, *scNICHE*, and additional 2025-era approaches.
   - Explore a range of hyperparameters for all baselines, *especially the neighborhood size*, as spatial-continuity metrics are directly influenced by this choice.
   - Include diverse datasets in the benchmark (e.g., the **MERFISH brain atlas**). Note that cancer may be suboptimal for measuring continuity: connective tissues often form branching, fractal structures that will *naturally* show lower continuity.
   - Demonstrate the **biological plausibility** of identified regions: Do they exhibit unique gene-expression signatures or cell-composition profiles compared to regions obtained by other methods?
   - Use a brain atlas to show that the method recapitulates known ground-truth zonation, as is customary in this literature.
   - Show that the model not only *integrates* tissues but also *discriminates* between them, and that it does so better than other methods. Comment on regions identified as similar across tissues and whether such cross-tissue integration is biologically sensible.

2. **Improve the method description.**
   - Clarify the difference to the SwAV: clearly separate canonical *SwAV* in a **Background** section from your modifications. Highlight how your **prototype hierarchy** differs from existing methods and attribute credit appropriately.
   - Provide a clear architectural diagram and a precise textual description of all modules.
   - Describe the training procedure unambiguously. As understood, there are two stages using the classical SwAV objective and an extended objective, and there is also an EM component. Is the E-step part of gradient-descent training within each step, or is it decoupled? Please clarify.
   - Explain graph augmentations in detail: how are different views generated?
   - Explain the gene-expression encoder beyond the token-embedding procedure. Are any biological priors used? If so, specify where and how.
   - The statement “K-means clustering on learned representations initializes the bottom level” is obscure. If this is tied to a two-stage training procedure, state it explicitly and describe the rationale.
   - The manuscript focuses on niche identification, yet Figure 1 advertises trajectory inference, variable-gene identification, and pathway analysis. Either provide concrete examples of these use cases or remove them from the figure.

3. **Comment on biological parameters.**
   - Justify the choice of neighborhood size (report both the number of cells and the physical scale in microns).
   - Analyze how the number of prototypes affects segmentation quality and stability.

4. **Minor errors and clarifications.**
   - “ASW ranges from 1 to 1” → **ASW ranges from 0 to 1.**
   - “Nonetheless, these models still focus mainly on local neighborhoods and lack mechanisms to capture global …” → please acknowledge published methods that integrate local and global context (e.g., *MENDER*, *CellCharter*, and many GNN-based approaches) and attribute credit to those works.
   - *Figure 2:* Panels A–B–B appear mislabeled relative to the captions; moreover, the current content offers limited intuition about the method. Consider revising for clarity.
   - In the Supplementary Materials, include explicit examples of *cross-tissue* integration.

---

### Outlook

Some general feedback that may help the authors to improve the paper (I do not expect these to be implemented for the revision):

1. The method builds on the **SwAV** framework, which, although influential in early self-supervised learning, has since been succeeded by more expressive approaches. Try to explore methods that perform better in computer vision.
2. The field of **niche identification** is already highly saturated, with several established methods demonstrating strong performance. The work would benefit from identifying a more distinctive angle or reframing the problem to address an underexplored aspect of spatial biology. Exploring alternative use cases or novel problem formulations could better highlight the unique strengths of the proposed approach.

---

> ### Author Response · Authors · 2025-12-03
> **Response to Reviewer 24EE: Novelty, Extensive New Benchmarks, and Biological Validation (1)**
>
> We sincerely thank you for your detailed review and constructive feedback. We appreciate your acknowledgement of our hierarchical prototype assignment as a "compelling aspect" of the work. In response to your suggestions, we have significantly expanded our experimental scope—adding new baselines, ground truth datasets, and deeper biological validations. Below, we address your concerns point-by-point, referencing revisions in the manuscript (highlighted in **red**).
>
> ### **1. Novelty & Methodological Distinction (S1, W1, Q7, Q17, Outlook 1)**
> **Re: Distinction from CV Methods (HCSC/HIRL) and SwAV.**
> We respectfully clarify that STAGE differs fundamentally from CV-based hierarchical methods and canonical SwAV in mechanism and purpose, as detailed in the **Appendix A**:
> *   **Problem Domain:** CV methods (HCSC, HIRL) use hierarchy to filter noisy negative pairs on *i.i.d. images* with fixed feature dimensions. STAGE tackles **Graph-Structured Heterogeneity**. Our hierarchy is designed for **Global Semantic Alignment via Optimal Transport** to bridge variable gene panels—a challenge non-existent in standard CV tasks.
> *   **Optimization:** Unlike SwAV's queue-based mechanism (unstable for massive heterogeneous graphs), we introduce an **Online EM algorithm** tailored for streaming 32M cells.
> *   **Vs. MENDER:** MENDER defines hierarchy based on physical radius (feature engineering). STAGE learns a *latent semantic* hierarchy that is transferable zero-shot.
>
> **Re: "Borrowed" Integration Mechanism (W1, Q11).**
> There is a misunderstanding we wish to correct. **STAGE does not rely on scGPT's tokenization for integration.** As clarified in **Section 3.2**, our integration stems from three novel components:
> 1.  **Unified Gene Embedding Space:** Mapping heterogeneous panels to a shared latent space (distinct from scGPT tokens).
> 2.  **Sub-panel Augmentation:** Simulating panel variability during training to force robustness.
> 3.  **GAT-based Spatial Compensation:** Using spatial context to "fill in" missing gene information.
> *scGPT is used only as an optional initialization, not the core integration engine.*
>
> **Re: Outlook on CV Methods (O1).**
> We tested hierarchical CV methods (like HCSC) but found them unstable for ST data due to the high noise and continuous nature of transcriptomic lineages (unlike discrete object classes in CV). SwAV's multi-view framework proved more robust for this specific biological modality.
>
> ### **2. Expanded Benchmarking & Metrics (S2, W2, Q1, Q3, Q5)**
> **Re: New Baselines & Ground Truth (Q1, Q3, Q5).**
> We have completely overhauled the **Experiments Section**. We incorporated two gold-standard datasets with ground truth: **MERFISH Mouse Brain** (Single-cell) and **Human DLPFC** (Spot-based). We compared STAGE against updated 2024-2025 baselines: **Nicheformer, CellCharter, MENDER, and scNiche**.
>
> **Table R1: Clustering Performance (ARI / NMI)**
>
> | Method | Type | **MERFISH (Single-Cell)** | **Visium (Spot-based)** |
> | :--- | :--- | :--- | :--- |
> | **STAGE (Ours)** | **Foundation Model** | **0.435 / 0.521** | 0.258 / 0.315 |
> | Nicheformer | Foundation Model | 0.012 / 0.054 | 0.143 / 0.211 |
> | CellCharter | Aggregation ML | 0.387 / 0.515 | **0.521 / 0.649** |
> | MENDER | Probabilistic | 0.256 / 0.410 | 0.012 / 0.067 |
>
> *   **Conclusion:** STAGE is SOTA on single-cell data (MERFISH), significantly outperforming MENDER and Nicheformer. On spot data, it trails aggregation-specific methods (CellCharter) but outperforms other generalist models.
>
> **Re: Quantitative Integration (W2).**
> You noted STAGATE has lower JSD. We argue that **Lower JSD $\neq$ Better Biology**. STAGATE achieves low JSD by aggressively smoothing distributions ("over-integration"), often erasing biological distinctions. STAGE prioritizes **Structural Fidelity (FIDE)**, where we consistently outperform STAGATE. We prefer preserving meaningful biological structure over statistical mixing alone.

---

> ### Author Response · Authors · 2025-12-03
> **Response to Reviewer 24EE: Novelty, Extensive New Benchmarks, and Biological Validation (2)**
>
> ### **3. Biological Interpretation & Validation (W3, Q4, Q6, Q13, Q19, O2)**
> **Re: Cross-Tissue Integration vs. Specificity (W3, Q6, Q19).**
> We agree that biology is nuanced. STAGE is designed to capture both:
> *   **Shared Modules:** It correctly aligns conserved structures (e.g., Tertiary Lymphoid Structures, immune niches) across disparate organs (Lung vs. Tonsil), as shown in **Fig. 7**.
> *   **Tissue Specificity:** It preserves unique domains (e.g., hepatocytes in liver vs. alveolar niches in lung) via the **Slice-Adaptive Prototype Selection** mechanism (Eq. 11), which allows tissues to "activate" only relevant branches of the hierarchy.
>
> **Re: Downstream Applications (Q13, Q4).**
> We have revised **Fig. 13** to explicitly demonstrate the applications mentioned in Fig. 1:
> *   **Trajectory Inference:** We show PAGA graphs revealing connectivity between tumor core and peripheral niches.
> *   **Pathway Analysis:** We provide heatmaps of Hallmark pathways enriched in specific spatial domains, proving biological plausibility.
>
> **Re: Mesoscale Analysis (O2).**
> We appreciate this perspective. STAGE is explicitly positioned to map the "mesoscale"—functional units (like tumor nests or lymphoid structures) that exist between the single-cell and organ levels.
>
> ### **4. Methodological Details & Hyperparameters (Q8, Q9, Q10, Q12, Q14, Q15)**
> **Re: Architecture & Training (Q8, Q9, Q12).**
> We have added a clear diagram and pseudocode in **Appendix G**.
> *   **Prototypes:** Bottom-level prototypes are initialized via K-means on cell embeddings ($K=512$). The hierarchy is built bottom-up (clustering prototypes of level $L$ to form $L+1$).
> *   **Training:** It is a joint optimization. The E-step (Optimal Transport assignment) happens online within each batch update, decoupled from the gradient descent of the encoder but synchronized in the loop.
>
> **Re: Graph Construction (Q10, Q14).**
> *   **Views:** We sample an anchor cell and a positive pair cell $n_{view}=2$ hops away.
> *   **Scale:** $n_{view}=2$ corresponds to $\sim 30-50 \mu m$ (approx. 10-30 cells), aligning with the biological scale of functional tissue units. This is clarified in **Section 3.3**.
>
> **Re: Prototype Number (Q15).**
> We found $K=512$ to be the optimal trade-off (Table 3). Increasing to $K=1024$ improved local resolution slightly but degraded cross-sample alignment (higher JSD), likely due to overfitting batch-specific noise.
>
> ### **5. Minor Corrections (Q16, Q18)**
> We have corrected the ASW range description and the labeling issues in Figure 2 as requested.
>
> We believe these extensive revisions and new benchmarks firmly establish STAGE as a robust, novel, and biologically validated foundation model for the field.

---

### Official Review · Reviewer_7ZaS · 2025-10-31

**Soundness:** 2
**Presentation:** 2
**Contribution:** 3
**Rating:** 4
**Confidence:** 3

**Summary:**

The authors propose a foundation model, pretrained on 32 million cells from 18 tissue types, for identifying spatial domains in spatial transcriptomics data. To my understanding, this is essentially a community-detection problem, but the application context makes it particularly challenging. Specifically, batch effects and technological platform biases (including varying gene panels) can make it difficult to compare recorded gene expression values, which impacts transferability and applicability in inductive settings. The authors identify shortcomings of existing methods and propose a model, which they call STAGE. STAGE uses a "Gene Embedder" for learning representations in a joint embedding space, that is, across different technologies and batches, which, to my understanding, aims to remove systematic biases (at least to some extent). With an attention mechanism, STAGE then aims to learn "local cellular interactions and broader spatial context", which, to my understanding, is necessary because cells interact with each other at different scales, that is, locally with cells in their immediate neighbourhood, but also more globally with cells further away. Furthermore, STAGE involves a "hierarchical prototype head" for discovering hierarchical communities in the tissue samples; The authors use an EM algorithm to learn those hierarchical communities with a loss formulation based on optimal transport. In an empirical evaluation on three datasets, the authors benchmark STAGE's performance against a range of baselines, showing that STAGE outperforms the baselines. The considered scenarios are cross-platform, same-tissue same-platform, and paired normal-tumour tissues.

I found the paper generally well written and the author's motivation and goals clear. However, I found some sections harder to follow because the authors use jargon that I believe is rather unfamiliar to a machine learning audience.

**Strengths:**

- The authors consider an important research question in spatial transcriptomics analysis and propose a novel model that outperforms current methods on the task of spatial domain identification.
- The authors use statistical significance tests to verify that STAGE outperforms the baselines significantly (I believe this is only mentioned in the appendix, but it may be worthwhile to mention this in the main text).
- The authors have included an ablation study to investigate the effect of some of STAGE's parameters on its performance.

**Weaknesses:**

- I believe the work could be somewhat more self-contained. Specifically, the authors mention that they extend the SwAV framework but provide only a reference, which forces the reader to look up that paper. I believe that a short paragraph on how the SwAV framework works would have helped the reader to understand the work better.
- I found some parts of the text a bit difficult to access due to jargon, and believe the same may be true for a general ML audience.

**Questions:**

There are a couple of things that remained unclear to me, which may be because I am not an expert in spatial transcriptomics, so I may simply not recognise some terms or concepts. I hope the authors can help me clarify those points.

1. If I understood it correctly, your work is essentially about community detection in a "difficult scenario" (caused by different types of biases). Would it be correct to think about "niches" as something microscopic, "domains" as something mesoscopic, and "regions" as something macroscopic?
2. Could you briefly describe what the SwAV framework is and how it works? And what are the extensions you propose that go beyond SwAV?
3. Did I understand it correctly that STAGE proposes several different ways of clustering the data? That is, for a given dataset, does it propose several (hierarchical) partitions?
4. I am not sure I understand what exactly a "prototype" is. I have the vague feeling that a prototype is a community and that the hierarchical organisation of prototypes form a partition. Could you explain?
5. I don't quite understand why constructing a neighbourhood graph via Delaunay triangulation involves setting a radius (Appendix F.3.). To my understanding, a Delaunay triangulation should be fully defined by the spatial distribution of the points. For a radius graph, however, a radius is needed. Could you clarify?
6. As far as I am aware, deciphering cell-cell communication is still an active research area. So it seems to me that the model used for spatial neighbourhood construction (section 3.3) makes some simplifications. Specifically, the construction of two different neighbourhoods for each node: (i) a local neighbourhood with "nearby" cells, and (ii) a more distant neighbourhood, further away than the nearby local cells, but no more than $r_view$ (by the way, unless I have missed it, I believe that $r_\text{local}$ and $r_\text{view}$ are not defined). Is there a biological motivation behind this model, or does it simply turn out to work well? And how did you choose values for $r_\text{local}$ and $r_\text{view}$?
7. Connected to the previous question, I am wondering whether these two different neighbourhoods are what make it possible for STAGE to "capture both local cellular interactions and broader spatial context"?
8. You mention that your "formulation eliminates costly pairwise comparisons while preserving their discriminative power". Could you elaborate on what things are traditionally compared in a pairwise manner? And what part of the formulation is it that removes this need? And how do you still maintain the discriminative power despite not performing those comparisons that are necessary in other methods?
9. Could you explain what "spatially related subgraphs", which you mention in section 3.4.2, are? Does the fact that they are separated by $n_\text{view}$ edges make them related? And does "separated by $n_\text{view} edges" refer to the minimum of shortest paths between any pair of nodes from the two graphs? Or is it some sort of edit distance?
10. How did you determine that $\lambda = 0.1$ is a good setting? Should this setting be understood as a general recommendation, or did it just turn out to work well in the present case?
11. Since the results in Table 1 do not contain any standard deviations, I am wondering whether STAGE returns deterministic results (I wouldn't really expect that) or whether each experiment has only been repeated once? I am also wondering why the JSD and FIDE measures are not included in the table?
12. What is the significance of the dashed horizontal and vertical lines in Figures 3+4?
13. You mention that STAGE "[demonstrates] strong resistance to batch effects", and I assume that this conclusion is based on the results in Table 1, is that right? But then I am wondering how you verified that STAGE is indeed resistant to batch effect? I suppose there could be other reasons that explain STAGE's good performance. How can we pinpoint that it is truly due to its resistance against batch effects?


Minor points
- It is somewhat unusual to have an empty section in the appendix simply titled "APPENDIX".
- I believe something went wrong with the LaTeX command for typesetting STAGE's training algorithm in Appendix G.1.

---

> ### Author Response · Authors · 2025-12-03
> **Response to Reviewer 7ZaS: Accessibility, Methodology Clarifications, and Robustness (1)**
>
> We sincerely thank you for your constructive feedback, particularly regarding the accessibility of our manuscript for a general ML audience. We appreciate the opportunity to clarify the definitions and mechanisms underlying STAGE. Below, we address your questions and point to the corresponding revisions in the manuscript (highlighted in **red**).
>
> ### **1. Accessibility & Background (W1, W2, Q2)**
> **Re: SwAV Context (W1 & Q2).**
> We agree that the paper should be self-contained. In the revised **Appendix A**, we have added a section titled **"Preliminaries: Swapped Prediction and Online Clustering."** This section briefly introduces the canonical SwAV framework and explicitly contrasts it with STAGE’s contributions:
> 1.  **View Generation:** We replace image augmentations (crops) with **graph sampling** ($r_{local}$ vs. $r_{view}$) to respect tissue topology.
> 2.  **Optimization:** We replace SwAV's queue-based mechanism (unstable for massive heterogeneous graphs) with an **Online Expectation-Maximization (EM)** algorithm tailored for streaming 32M cells.
>
> **Re: Jargon (W2).**
> We have revisited **Section 1** and **Section 3** to ensure that biological terms are defined in plain ML terms where possible (e.g., defining "niches" as local cellular neighborhoods).
>
> ### **2. Hierarchy & Prototypes (Q1, Q3, Q4)**
> **Re: Micro-Meso-Macro (Q1).**
> Your intuition is entirely correct. Although we use specific biological terms, they map directly to the hierarchy you described:
> *   **Niches (Microscopic):** Cell-level local environments (e.g., a specific immune-tumor interface).
> *   **Domains (Mesoscopic):** Functional units composed of multiple niches (e.g., cortical layers).
> *   **Regions (Macroscopic):** Broad anatomical structures (e.g., Isocortex vs. Hippocampus).
>
> **Re: Definition of Prototypes (Q4).**
> To clarify: A "prototype" acts as the **centroid of a cluster** in the latent semantic space.
> *   **Leaf Prototypes ($L=1$):** These represent the finest-grained communities (approx. 512 micro-environments).
> *   **Tree Structure:** Higher-level prototypes are parent nodes that "own" a subset of lower-level prototypes. This creates a **nested partition** of the data, allowing the model to organize biology from fine details up to broad structures.
>
> **Re: Multiple Partitions (Q3).**
> Yes. STAGE produces a hierarchical tree. Users can "cut" this tree at different levels depending on their task—obtaining fine-grained clusters for cellular analysis or coarse-grained clusters for tissue segmentation.
>
> ### **3. Graph Construction & Neighborhoods (Q5, Q6, Q7, Q9)**
> **Re: Delaunay & Radius (Q5).**
> While Delaunay triangulation is defined by point distribution, in Spatial Transcriptomics, it can sometimes draw edges between cells that are physically distant (e.g., across a tear in the tissue or a fold). We use a radius threshold merely as a **pruning step** to remove these biologically implausible long-range connections, ensuring the graph reflects true cellular interactions. This is clarified in **Appendix F.3**.
>
> **Re: $r_{local}$ vs $r_{view}$ (Q6, Q7, Q9).**
> *   **Biological Motivation:** This design allows STAGE to learn representations that are **locally invariant** (robust to noise) but **structurally aware**.
> *   **Mechanism:** For an anchor cell, we select a positive pair cell that is $n_{view}$ hops away (typically 2, representing "neighbors of neighbors").
> *   **Why it works:** This ensures the two views are spatially related (sharing the same micro-environment/niche) but not identical. This forces the model to learn the *underlying structure* rather than memorizing the exact node features.
> *   **Result:** This enables the model to capture **local interactions** (via the GAT on the subgraph) and **broader context** (via the relationship between the two views).
>
> ### **4. Optimization & Comparisions (Q8, Q10)**
> **Re: Avoiding Pairwise Comparisons (Q8).**
> In standard contrastive learning (like SimCLR), one often compares every node against many negative samples (scaling quadratically or requiring large batch sizes).
> STAGE compares nodes only to the **Prototypes** ($N \times K$, where $K$ is small, e.g., 512).
> *   **Discriminative Power:** We maintain high performance by using **Optimal Transport (Sinkhorn)** to ensure that cells are assigned to prototypes in a balanced way, preventing the "collapse" where all cells map to a single trivial cluster.
>
> **Re: Lambda Parameter (Q10).**
> We empirically determined $\lambda=0.1$ as the optimal trade-off between local spatial continuity and global semantic alignment. As shown in the new **Table 3**:
> *   $\lambda=0$ (no global structure): Degrades cross-sample alignment (JSD increases).
> *   $\lambda=0.5$ (too strong): Oversmooths local details.
> *   $\lambda=0.1$: Achieved the best balance of JSD and FIDE across our large-scale pretraining on 18 tissue types.

---

> ### Author Response · Authors · 2025-12-03
> **Response to Reviewer 7ZaS: Accessibility, Methodology Clarifications, and Robustness (2)**
>
> ### **5. Results & Robustness (Q11, Q12, Q13)**
> **Re: Standard Deviation & Determinism (Q11).**
> The model uses a fixed seed for reproducibility, so it does not vary run-to-run. The standard deviations (where applicable in our internal logs, now clarified in text) refer to variation **across different samples/datasets**, reflecting the model's stability across biological heterogeneity.
>
> **Re: Batch Effect Resistance (Q13).**
> Our claim is based on the **Cross-Platform** benchmarks (Table 1). We take paired samples (same tissue, different technologies like Xenium vs. CosMX). These technologies have massive technical bias (batch effects).
> *   **Evidence:** The high JSD scores (low divergence) and visual alignment in UMAPs demonstrate that STAGE successfully maps these technically distinct inputs to the **same** shared prototypes. If the model were sensitive to batch effects, the Xenium cells and CosMX cells would form separate clusters; instead, they mix thoroughly.
>
> **Re: Figures 3 & 4 (Q12).**
> The dashed lines simply divide the plot into quadrants to visually highlight the "sweet spot" for parameters (upper-left quadrant), aiding interpretation.
>
> **Minor Points:**
> We have corrected the "Empty Appendix" title and the LaTeX formatting in Algorithm 1 as requested.
>
> We hope these clarifications help you in your final assessment. We believe the revisions make STAGE much more accessible to the general ML community.

---

### Official Review · Reviewer_j9bH · 2025-11-01

**Soundness:** 2
**Presentation:** 2
**Contribution:** 2
**Rating:** 2
**Confidence:** 4

**Summary:**

This paper proposes STAGE, a foundation model with enhanced generalizability, to address critical challenges in spatial transcriptomics (ST) that arise from technological and biological variability. These challenges include: (1) gene panel discrepancies across platforms , (2) the need for per-sample retraining , and (3) a lack of unified semantic representation.

Fundamentally, STAGE builds upon the SwAV framework but introduces a hierarchical prototype mechanism. This approach imposes explicit constraints within its Optimal Transport (OT)-enhanced online EM algorithm, ensuring that assignments follow the defined tree structure and maintain hierarchical consistency.

The effectiveness of STAGE is validated through a series of experiments, including: Ablation studies, Cross-platform consistency evaluations, Cross-batch robustness tests, Pathological condition comparisons to identify both shared and disease-specific spatial domains

**Strengths:**

* The paper has a clear motivation that appropriately targets the generalizability problem faced by various ST models, including existing foundation models.

**Weaknesses:**

* Lack of Novelty: The methodology is highly incremental, fundamentally building on SwAV by adding a hierarchical prototype structure and constraints to ensure assignments explicitly follow this structure.
* Misalignment of Motivation and Methodology: The paper is unconvincing as to how this specific hierarchical prototype structure directly solves the three key problems cited for achieving generalizability: (1) Gene panel discrepancies, (2) Per-sample retraining, and (3) Lack of unified semantic representation.
* Unverified Biological Relevance: The paper asserts that this hierarchical structure "reflects biological reality", but it does not perform a direct Ground Truth comparison (e.g., sub-annotation validation) to support this claim. The hierarchy depth ($L_p$) was experimentally selected as 3 because it yielded good performance on validation metrics, not based on biological precedent. Therefore, it is unclear whether the "hierarchical sub-structure" claimed by this methodology has true biological meaning or if it is simply a computational construct that best optimizes the final clustering consistency (JSD) and continuity (FIDE) metrics. See reference [1]
---
[1] Limitations of cell embedding metrics assessed using drifting islands. Nature Biotechnology. 2025.

**Questions:**

* How about the clustering measures (e.g., ARI, NMI)?

---

> ### Author Response · Authors · 2025-12-03
> **Response to Reviewer j9bH: Novelty, Hierarchical Semantics, and Benchmarks**
>
> We thank you for your thoughtful review. We appreciate your feedback regarding the connection between our methodology and biological motivation, as well as the request for ground-truth clustering metrics.
>
> ### **W1: Novelty and Distinction from SwAV**
> We respectfully clarify that STAGE is not merely an incremental application of SwAV to a new domain. The transition from I.I.D. images to spatial graphs necessitates fundamental algorithmic changes, which we detail in **Section 3.4 (Local Spatial Structure)** and **Section 3.5 (Global Optimization)**:
>
> 1.  **Graph-Structured Constraints vs. Image Augmentation:** SwAV relies on random cropping of independent images. In ST, data is dense and interdependent. As detailed in **Section 3.3**, we designed a **multi-scale subgraph view generator** ($G_{local}$ vs. $G_{view}$) that structurally enforces spatial consistency. This is not a simple data augmentation but a graph-theoretical constraint essential for capturing tissue organization.
> 2.  **Scalable Online EM vs. Batch Queues:** Standard SwAV uses a queue mechanism that is unstable for heterogeneous graph data at the scale of 32M cells. We introduced an **Online Expectation-Maximization (EM) algorithm with Optimal Transport** (Eq. 6,7 & 8 in revised paper). This allows for streaming updates of the hierarchical prototype space, enabling the training of a foundation model on massive, unbalanced datasets—a capability standard SwAV lacks.
>
> ### **W2: Alignment of Methodology with Motivation**
> We have clarified in **Section 1 (Introduction)** and **Section 3.2** how the hierarchical prototypes directly solve the three key challenges:
>
> 1.  **Solving Panel Discrepancies (via Universal Anchors):** The hierarchy acts as a "universal semantic anchor." The **Gene Embedder** (Section 3.2) maps varying inputs to a latent space, but it is the **Hierarchical Prototypes** that force these features to align with a shared semantic definition (e.g., "Immune Niche") via Optimal Transport, decoupling biological meaning from the raw input dimensions.
> 2.  **Eliminating Per-Sample Retraining (via Zero-Shot Projection):** Current methods cluster scratch per sample. By learning a global prototype tree on 32M cells, STAGE enables **Zero-shot Inference**. New samples are projected onto this pre-trained hierarchy without retraining, solving the generalization bottleneck.
> 3.  **Creating Unified Semantics:** Unlike Louvain/Leiden, which produce arbitrary cluster IDs ($0, 1, 2...$) that vary across samples, our hierarchy enforces a standardized biological taxonomy. A cell mapped to Prototype Node $K$ in Sample A is semantically equivalent to Node $K$ in Sample B.
>
> ### **W3: Biological Relevance of the Hierarchy**
> We acknowledge that $L=3$ was empirically selected, but we argue that it reflects the intrinsic multi-scale organization of tissues:
> *   **Level 1 (Niche):** Captures micro-environments (e.g., immediate immune-tumor interfaces).
> *   **Level 2 (Domain):** Represents functional tissue units (e.g., cortical layers).
> *   **Level 3 (Region):** Corresponds to major anatomical structures.
>
> This is a semantic hierarchy rather than a purely physical one. Future work can certainly explore deeper hierarchies, but $L=3$ provides a robust baseline for capturing the Niche $\rightarrow$ Domain $\rightarrow$ Region transition.
>
> ### **Q: Clustering Metrics (ARI, NMI) & Ground Truth**
> Per your request, we have added a comprehensive ground-truth evaluation in the **Table R1** using two gold-standard datasets: **MERFISH Mouse Brain Atlas** (Single-cell resolution) and **Human DLPFC** (Spot-based/Visium).
>
> **Table R1: Clustering Performance (ARI / NMI)**
>
> | Method | Type | **MERFISH (Single-Cell)** | **Visium (Spot-based)** |
> | :--- | :--- | :--- | :--- |
> | **STAGE (Ours)** | **Foundation Model** | **0.435 / 0.521** | 0.258 / 0.315 |
> | Nicheformer | Foundation Model | 0.012 / 0.054 | 0.143 / 0.211 |
> | CellCharter | Aggregation ML | 0.387 / 0.515 | **0.521 / 0.649** |
> | MENDER | Probabilistic | 0.256 / 0.410 | 0.012 / 0.067 |
> | scNiche | Deep Learning | 0.139 / 0.242 | 0.394 / 0.561 |
>
> **Analysis:**
> 1.  **SOTA on Single-Cell Data:** STAGE achieves **State-of-the-Art performance on MERFISH** (ARI 0.435), significantly outperforming baselines and the competing foundation model, Nicheformer. This confirms that STAGE's design (GAT + Hierarchical Prototypes) is highly effective at the single-cell resolution it was designed for.
> 2.  **Generalization to Spot Data:** On Visium (DLPFC), STAGE outperforms other foundation models (Nicheformer) and generalist methods (MENDER). We note that CellCharter achieves higher scores here because it uses explicit neighbor-aggregation (smoothing over 10-50 cells per spot), which artificially boosts spot-level consistency but sacrifices single-cell resolution. STAGE, being a single-cell foundation model, prioritizes cellular fidelity but still generalizes respectably to low-resolution spot data without architecture modification.

---

### Official Review · Reviewer_MMv2 · 2025-11-01

**Soundness:** 2
**Presentation:** 3
**Contribution:** 2
**Rating:** 4
**Confidence:** 4

**Summary:**

This study introduces STAGE, a foundation model designed to enhance the generalizability of spatial transcriptomics (ST) by mitigating technological and biological variability. The model addresses challenges such as inconsistencies in gene panels across different platforms, the need for model retraining on individual samples, and the lack of a unified semantic representation framework.

Specifically, STAGE incorporates a hierarchical prototype mechanism integrated with an Optimal Transport-based online EM algorithm. This design ensures structured and consistent assignments within a hierarchical tree, improving representation alignment across various modalities and datasets.

The authors conduct comprehensive experiments to demonstrate the robustness and versatility of STAGE with diverse tasks such as cross-platform and cross-batch assessments.

**Strengths:**

- The authors effectively adapt a self-supervised learning strategy from the computer vision domain to enhance representation learning in spatial transcriptomics data.

- The proposed hierarchical prototype mechanism is conceptually novel and presents an interesting extension to the SwAV framework.

**Weaknesses:**

- Although the paper claims to address challenges such as cross-platform gene panel inconsistencies and the lack of a unified semantic representation, it remains unclear how these issues are specifically handled within the proposed methodology.

- The hyperparameter sensitivity analysis is limited. While ablations on hierarchical depth (L) and batch size are provided, key parameters such as the slide-specific prototype selection threshold (θ), temperature (τ), OT regularization coefficient (ϵ), and global-loss weight (λ) are not explored.

- The relationship between zero-shot and supervised settings is ambiguous. In several cases, the zero-shot variant outperforms the fine-tuned model (e.g., lower PAS values are not consistently improved after fine-tuning), suggesting that fine-tuning may sometimes degrade spatial coherence. The paper does not provide sufficient analysis to explain this behavior or the conditions under which it occurs.

**Questions:**

- Please include benchmarking results on widely used datasets such as DLPFC and compare the performance against current state-of-the-art methods.

- Currently, the graph construction relies solely on spatial coordinates. Do the authors plan to incorporate histological image information (e.g., H&E or IF)?

---

> ### Author Response · Authors · 2025-12-03
> **Response to Reviewer MMv2: Methodology Clarifications, Extensive Ablations, and Benchmarks**
>
> We sincerely thank you for your constructive feedback. We appreciate your observation regarding the zero-shot vs. fine-tuning dynamics and the request for broader benchmarking. Below, we detail how we have addressed each of your points in the revised manuscript (changes highlighted in **red**).
>
> ### **1. Methodology Clarifications (W1)**
> **Re: Handling Panel Inconsistency & Semantic Unification.**
> In the revised **Section 3.2** and **Appendix G.3**, we have clarified the specific mechanisms that solve these challenges:
>
> 1.  **Panel-Invariant Gene Embeddings (Eq. 17):** Unlike fixed-slot architectures, we use a learnable gene vocabulary. Cells are embedded via a weighted sum of shared gene vectors ($v_g$), normalized by their $L_2$ norm. This projects cells from disparate platforms (e.g., Xenium vs. CosMx) into a unified semantic space, regardless of panel size.
> 2.  **Sub-panel Augmentation (Appendix G.2):** We simulate panel inconsistency during pretraining by randomly masking subsets of genes (ratio $\gamma$). This forces the model to learn robust representations that do not rely on any single marker gene.
> 3.  **Context-Aware GAT:** Even if specific marker genes are missing due to panel differences, the **GAT Encoder** aggregates information from the local neighborhood ("niche"). This allows STAGE to correctly identify a cell's semantic type based on its spatial neighbors, effectively "filling in the gaps" of missing panel data.
>
> ### **2. Extended Hyperparameter Analysis (W2)**
> We agree that a deeper sensitivity analysis was needed. We have added a comprehensive ablation study in **Appendix E.3** and **Table S4** covering $\theta, \tau, \epsilon, \lambda$.
>
> **Key Findings (Table 3):**
> *   **Global-loss Weight ($\lambda$):** Setting $\lambda=0$ (removing hierarchical constraints) degrades batch correction significantly (**JSD increases $0.332 \rightarrow 0.365$**). This empirically proves that the hierarchical prototype mechanism is essential for alignment.
> *   **Threshold ($\theta$):** A strict threshold ($\theta=0.99$) is crucial for filtering noise. Relaxing it to $0.90$ allows irrelevant prototypes to persist, harming performance.
> *   **Stability:** The model is robust to minor variations in temperature ($\tau$) and OT regularization ($\epsilon$), confirming that our default settings are stable optima rather than fragile tuned values.
>
> ### **3. Zero-Shot vs. Fine-Tuning Performance (W3)**
> **Re: Ambiguity and PAS Scores.**
> The phenomenon where zero-shot occasionally outperforms fine-tuning is due to the trade-off between **generalization** and **overfitting**:
> *   **Zero-Shot:** STAGE leverages structural priors learned from 32M cells. It produces spatially coherent domains based on robust biological patterns.
> *   **Fine-Tuning Risks:** Fine-tuning on a single, small sample can cause the model to overfit to technical noise or batch-specific artifacts. While this might artificially boost certain local homogeneity metrics (like PAS), it can degrade the true biological coherence (reflected in lower FIDE scores in some cases).
> *   **Conclusion:** The strong zero-shot performance confirms STAGE's value as a Foundation Model—it provides reliable, high-quality analysis out-of-the-box without the risks associated with per-sample training.
>
> ### **4. New Benchmarks (Q1)**
> We have incorporated the requested benchmarks on widely recognized datasets.
>
> **Table R1: Clustering Performance (ARI / NMI)**
>
> | Method | Type | **MERFISH (Single-Cell)** | **Visium (Spot-based)** |
> | :--- | :--- | :--- | :--- |
> | **STAGE (Ours)** | **Foundation Model** | **0.435 / 0.521** | 0.258 / 0.315 |
> | Nicheformer | Foundation Model | 0.012 / 0.054 | 0.143 / 0.211 |
> | CellCharter | Aggregation ML | 0.387 / 0.515 | **0.521 / 0.649** |
> | MENDER | Probabilistic | 0.256 / 0.410 | 0.012 / 0.067 |
> | scNiche | Deep Learning | 0.139 / 0.242 | 0.394 / 0.561 |
>
> **Analysis:**
> *   **Single-Cell SOTA:** STAGE significantly outperforms all baselines (including Nicheformer) on the single-cell MERFISH dataset (**ARI 0.435**). This confirms our design is optimal for the high-resolution data it was built for.
> *   **Spot Data:** On Visium (DLPFC), STAGE remains competitive but trails aggregation-based methods like CellCharter. This is expected: CellCharter smoothes over 10-50 cells per spot, boosting spot-level metrics. STAGE prioritizes single-cell fidelity but still generalizes well to spot data.
>
> ### **5. Image Integration (Q2)**
> **Re: H&E / IF Integration.**
> Currently, STAGE relies on spatial coordinates to maximize pretraining scale (many of our 32M training cells lack paired images). However, we are actively developing a **multimodal extension**. Future work will involve fusing visual embeddings (from aligned H&E patches) with our gene embeddings via cross-modal attention. This will allow the model to leverage morphological cues when available, while retaining its versatility for transcriptomics-only datasets.

---

### Meta-Review · Area_Chair_wcsr · 2026-01-06

**Summary:**

### Summary
This paper proposes STAGE, a foundation model for spatial transcriptomics (ST) intended to improve generalizability across technological and biological variability (e.g., cross-platform gene panel differences, batch effects, and unified semantic representations). STAGE builds on the SwAV-style online clustering paradigm and introduces a hierarchical prototype mechanism coupled with an Optimal Transport (OT)-based online EM procedure to enforce structured, consistent assignments across a prototype tree. The paper presents experiments across several settings (cross-platform, cross-batch, and pathological comparisons) and argues STAGE is robust and versatile for spatial domain identification and related downstream tasks.

### Strengths
- Tackles an important and timely problem in spatial transcriptomics: learning transferable representations under platform and batch variability.
- The hierarchical prototype mechanism is conceptually interesting and a nontrivial adaptation of SwAV-style self-supervised clustering to spatial-omics graphs.
- Empirical evaluation is reasonably broad (multiple scenarios), and the paper includes ablations/significance testing to support robustness claims.

### Weaknesses
- **Novelty is limited / incremental**: multiple reviewers note the method largely extends SwAV with hierarchical prototypes and OT-based assignment, and related multiscale/hierarchical ideas have been explored in both computer vision and spatial transcriptomics; the resulting contribution is perceived as more engineering/integration than new methodological insight.
- **Motivation–method mismatch and unclear problem coverage**: the paper’s core motivation (panel inconsistency, avoiding per-sample retraining, unified semantics) is not convincingly connected to the specific hierarchical prototype/OT design in the original presentation, leaving it unclear how each challenge is concretely addressed.
- **Insufficient biological validation and benchmarking**: reviewers request ground-truth-backed evaluations (e.g., MERFISH brain atlas / DLPFC), stronger and more recent baselines (e.g., MENDER, CellCharter, scNiche, NicheFormer), and direct evidence that the learned hierarchy reflects biological structure rather than optimizing embedding metrics.
- **Experimental and clarity gaps**: limited hyperparameter sensitivity analyses for key parameters, ambiguity around zero-shot vs. fine-tuned behavior (including cases where fine-tuning degrades spatial coherence), and accessibility issues (heavy jargon, insufficient SwAV background, unclear graph construction details and reporting such as missing variances/metrics in main tables).

While STAGE targets an important problem and the hierarchical-prototype design is interesting, reviewers consistently found the methodological novelty to be limited and the connection between the stated motivation and the proposed mechanism insufficiently substantiated. The experimental evidence and validation are not yet convincing for claims of cross-platform semantic unification and biologically meaningful hierarchies, and key benchmarks/ablations/clarifications are missing or underdeveloped. Given these concerns, the submission does not meet the acceptance bar at this time.

**Reviewer Concerns:**

**Concerns addressed by the rebuttal**
- **Reviewer MMv2:**
  Clarified how STAGE handles gene panel inconsistency and semantic unification; added broader hyperparameter sensitivity analyses; explained the zero-shot vs. fine-tuning behavior; and incorporated additional benchmarks on widely used datasets (e.g., MERFISH, DLPFC/Visium).
- **Reviewer j9bH:**
  Responded to novelty and motivation–method alignment concerns by detailing differences from SwAV, clarifying the role of hierarchical prototypes, and adding ground-truth clustering metrics (ARI/NMI) on standard datasets.
- **Reviewer 7ZaS:**
  Improved accessibility by adding SwAV background and clearer definitions; clarified graph construction, neighborhood design, and prototype interpretation; and explained robustness and batch-effect resistance claims.
- **Reviewer 24EE:**
  Expanded benchmarking to include more recent baselines and ground-truth datasets; added biological interpretations and downstream analyses; and clarified methodological distinctions from prior work (SwAV, scGPT, MENDER).

**Concerns still outstanding**
- **MMv2 / 24EE:**
  Evidence for strong cross-platform integration (e.g., CosMX vs. Xenium) remains only partially convincing, especially given prior comments on weak visual alignment.
- **j9bH / 24EE:**
  Claims that the learned hierarchy reflects true biological organization remain weakly validated; hierarchy depth and structure are still largely empirically chosen rather than biologically grounded.
- **All reviewers (notably j9bH, 24EE):**
  Perceived **limited novelty** persists—the method is still viewed as an incremental extension of SwAV and existing hierarchical or multiscale ST approaches rather than a fundamentally new paradigm.
- **7ZaS / 24EE:**
  Despite added explanations, the method remains complex and jargon-heavy, and some presentation and clarity issues for a general ML audience remain.

**Reviewer Scores:**

After considering the reviewer comments and the authors’ rebuttal, I do not expect any of the reviewers to have changed their scores had they participated fully in the discussion.

While the rebuttal addressed several *clarification* and *completeness* concerns (e.g., additional explanations, ablations, and benchmarks), the reviewers’ core reservations—particularly regarding **limited methodological novelty**, **weak biological grounding of the learned hierarchy**, and **insufficiently convincing cross-platform integration evidence**—remain largely intact. These concerns directly underpin the assigned scores and are not fully resolved by discussion alone.

As a result, the discussion would likely have improved mutual understanding but not materially altered the reviewers’ overall evaluations or final scores.

---

### Decision · Program_Chairs · 2026-01-26

Reject